# The "ABC model": a non-hydrostatic toy model for use in convective-scale data assimilation investigations

Ruth Elizabeth Petrie[1,2], Ross Noel Bannister[1], and Michael John Priestley Cullen[3]

[1]Dept. of Meteorology, Univ. of Reading, Earley Gate, Reading, RG6 6BB, UK
[2]Now at The Centre for Environmental Data Analysis, RAL Space R25 - Room 1.22, STFC Rutherford Appleton Laboratory, Harwell Oxford, Didcot, OX11 0QX, UK
[3]Met Office, FitzRoy Road, Exeter, EX1 3PB, UK

*Correspondence to:* Ross Bannister, r.n.bannister@reading.ac.uk

**Abstract.** In developing methods for convective-scale data assimilation (DA) it is necessary to consider the full range of motions governed by the compressible Navier-Stokes equations (including non-hydrostatic and ageostrophic flow). These equations describe motion on a wide range of time-scales with non-linear coupling. For the purpose of developing new DA techniques that suit the convective-scale problem it is helpful to use so-called 'toy models' that are easy to run, and contain the same types of motion as the full equation set. Such a model needs to permit hydrostatic and geostrophic balance at large-scales, but to allow imbalance at small-scales, and in particular, it needs to exhibit intermittent convection-like behaviour. Existing 'toy models' are not always sufficient for investigating these issues.

A simplified system of intermediate complexity derived from the Euler equations is presented, which supports dispersive gravity and acoustic modes. In this system the separation of time scales can be greatly reduced by changing the physical parameters. Unlike in existing toy models, this allows the acoustic modes to be treated explicitly, and hence inexpensively. In addition, the non-linear coupling induced by the equation of state is simplified. This means that the gravity and acoustic modes are less coupled than in conventional models. A vertical slice formulation is used which contains only dry dynamics. The model is shown to give physically reasonable results, and convective behaviour is generated by localised compressible effects. This model provides an affordable and flexible framework within which some of the complex issues of convective-scale DA can later be investigated. The model is called the "ABC model" after the three tunable parameters introduced: $A$ (the pure gravity wave frequency), $B$ (the modulation of the divergent term in the continuity equation), and $C$ (defining the compressibility).

## 1  Introduction

Advances in computer power have enabled Numerical Weather Prediction (NWP) models to operate at higher resolutions than has previously been possible. In 2009 the Meteorological Office (Met Office) upgraded the resolution of its Unified Model (UM, Davies et al. (2005)) for the UK domain from $12\,\mathrm{km}$ to $1.5\,\mathrm{km}$ (Dixon et al., 2009). Resolutions of this degree are expected to resolve the large and synoptic scale features well. Bryan et al. (2003) found that models with resolutions of $100\,\mathrm{m}$ are necessary to provide meaningful simulations of convection. Resolutions of $\mathcal{O}(100\,\mathrm{m})$ are not yet affordable over the UK domain with current computer resources, although research experiments with the UM over smaller domains with $200\,\mathrm{m}$

resolution have shown marked benefit (Lean et al., 2008). Models of $\mathcal{O}(\lesssim 1\,\text{km})$ resolution are known as convective-scale models because they are capable of resolving some convection explicitly, thus do not require a full convection scheme. For instance it is possible to explicitly represent features such as thunderstorms $\mathcal{O}(10\,\text{km})$ and Mesoscale Convective Systems (MCSs) $\mathcal{O}(10\text{-}100\,\text{km})$, though not necessarily resolve their internal structure (e.g. Bryan et al. (2003); Clark et al. (2005); Lean et al. (2008)). Convective-scale forecasting can facilitate more accurate and earlier indications of extreme or hazardous weather, e.g. severe convection (Lean et al., 2008), which is of clear benefit.

As NWP moves towards the convective-scale so it is appropriate to examine the data assimilation (DA) scheme underpinning the forecast. The DA process combines meteorological data from a variety of sources including satellites, radar, weather stations, and radiosondes with a previous forecast (a background state) to produce an analysis. The NWP model is then integrated forward from the analysed state. The DA scheme that combines the observed and background data should provide an analysis that is consistent with the observations and the model broadly within the specified 'bounds' of the observation errors, of the background (prior) state errors, and of the model errors (if a model error scheme is incorporated into the DA system). The development of our toy model is a step towards a detailed and technical investigation of the convective-scale DA problem though its utility is not limited to this application.

Convective-scale DA introduces new issues. The errors in the larger-scale flow are still present, but in addition there will be errors on the small scales resolved by the convective-scale model which will have a different correlation structure. A pragmatic solution is to rely on a larger scale DA system to correct the large-scale errors, and thus allow convective-scale DA to focus on small scales. The model introduced in this paper is intended to allow development of methods of assimilating information over all scales. Detailed reviews of the issues are given by Park and Županski (2003); Dance (2004); Sun (2005); Lorenc and Payne (2007).

The current Met Office operational large-scale DA scheme enforces hydrostatic balance as a strong constraint and exploits geostrophy as a weak constraint in the background error covariance model (Lorenc et al., 2000; Bannister, 2008). The use of the hydrostatic balance relationship is valid for flows where the aspect ratio is much less than one, e.g. Holton (2004); Vallis (2006). In regions of convection the aspect ratio increases and so hydrostatic balance may no longer be a good approximation. Vetra-Carvalho et al. (2012) demonstrated that hydrostatic balance breaks down in the UM when it is run at $1.5\,\text{km}$ horizontal resolution in regions of convection. At mid and high latitudes the geostrophic assumption is accurate for large-scale flows where the Rossby number is small (e.g. Holton (2004)). At the convective-scale the Rossby number is not small and therefore the use of geostrophic balance is no longer appropriate. It is therefore important that these balances are relaxed in convective-scale DA. Some variational DA methods, such as those termed "EnsVar" (Lorenc, 2013; Liu and Xue, 2016; Bannister, 2017) use information from an ensemble to represent background error covariance information without, in principle, the need to impose balances via a prescribed background error covariance matrix. These methods though suffer from noise in the sampled error covariance matrix and so rely on methods to mitigate its effect (namely by localisation, which is known to destroy balances when they are relevant (Kepert, 2009; Bannister, 2015)). The sampled (and localised) error covariance matrix in these methods is often hybridised with a prescribed background error covariance matrix (Clayton et al., 2013), which does impose balances.

This brings attention back to the validity of such balances when such methods are applied at convective-scales, and hence to simplified systems where this issue can be studied closely.

Operational systems have to resolve features at both the synoptic and the convective-scales, requiring a large number of grid points. Such systems are very expensive to run and are not ideal tools for research purposes. The wide range of time-scales means that semi-implicit integration schemes are required for efficiency, e.g. Davies et al. (2005), and the nonlinear coupling between acoustic and gravity waves through the equation of state makes analysing the small-scale behaviour difficult (Thuburn et al., 2002). Thus it would be useful to have a simplified model which describes a variety of regimes but without the extreme separation of time-scales and the full nonlinear coupling between acoustic and gravity waves present in the real system. A simplified system that has these properties allows problems such as the convective-scale DA problem to be explored in a practical but physically realistic way.

Perhaps the simplest non-linear model of convection is the well-known Lorenz 63 system (Lorenz, 1963), which describes convection with only three variables. These are (respectively) proportional to the strength of the convective motion, the size of the temperature differences between the up- and down-welling air, and the degree of deviation from linearity of the temperature profile. The resulting three ordinary differential equations are easily integrated numerically, but they miss the representation of the complex spatial aspects of the problem required to mirror real forecasting problems. Würsch and Craig (2014) discuss the lack of availability of suitable simplified models of convection for DA research, and they note that people have tended to run full NWP models for this purpose, but in idealised settings (see references in Würsch and Craig (2014) for examples). These models however remain complicated and expensive to run. Würsch and Craig (2014) developed a simplified model for purposes of convective-scale DA research. Their model is based on the one-dimensional shallow water model, modified to account for the phase transitions of cloud formation and precipitation – essential processes in the formation of cumulus convection. Although their model has shown to be very useful for this purpose, its one-dimensionality makes it impossible to tackle questions relating to the breakdown of hydrostatic balance, and to simulate our inability in practical situations to resolve vertical structures from observations.

The simplified system derived in this paper is intended to be run in vertical slice geometry (longitude/height), so that many fewer degrees of freedom are needed than in an operational three-dimensional system. The equations are modified so that the speed of the acoustic and gravity waves can be controlled, and so the normally large separations in time-scales can be reduced. The equation of state is also modified so that the degree of coupling between the acoustic and gravity waves is reduced. The modifications are designed so that energy is conserved in the equations, which is necessary for realistic behaviour. In order to study the dynamically-related breakdown of balance, no moisture is included, but intermittent convection-like behaviour is still seen (e.g. via gravity wave breaking). These simplifications permit large-scale balanced flows and sporadic small-scale non-hydrostatic flows (including convection) to coexist within the framework of a simplified and practical model.

The structure of this paper is as follows: section 2 provides a derivation of the toy model equations which are analysed in terms of a scale analysis and energy conservation properties. Section 3 describes the numerical implementation of the model. Section 4 provides a linear analysis of the equations. Section 5 shows the results of an idealized integration which illustrates how the model can be used to simulate different flow regimes. Section 6 provides a summary and some concluding remarks.

Future work will exploit this model in testing different approaches to convective-scale DA, as piloted in Petrie (2012). For reference, Table 1 summarises the symbols used throughout this paper.

## 2 Derivation of the model equations

The model is derived from the compressible 3-D Euler equations (1), see e.g. Holton (2004); Pielke (2001); Vallis (2006):

$$\frac{\partial \mathbf{u}}{\partial t} + \mathbf{u} \cdot \nabla \mathbf{u} + \frac{1}{\rho} \nabla p + g\mathbf{k} + f\mathbf{k} \times \mathbf{u} = 0, \tag{1a}$$

$$\frac{\partial \rho}{\partial t} + \nabla \cdot (\rho \mathbf{u}) = 0, \tag{1b}$$

$$\frac{\partial \theta}{\partial t} + \mathbf{u} \cdot \nabla \theta = 0, \tag{1c}$$

$$p = \rho R \left( \frac{p}{p_{00}} \right)^\kappa \theta. \tag{1d}$$

Equations (1a) are the momentum equations, where $t$ is time, $\mathbf{u} = (u, v, w)$ comprises zonal ($u$), meridional ($v$) and vertical ($w$) components, $p$ is pressure, $g$ is the acceleration due to gravity and $\rho$ is density. The $f$-plane assumption is made and $\mathbf{k}$ is the vertical unit vector. Equation (1b) is the compressible mass continuity equation. Equation (1c) is the adiabatic thermodynamic equation where $\theta$ is potential temperature. Equation (1d) is the equation of state where $p_{00} = 1000$ hPa, $\kappa = R/c_p$ is a constant, with $c_p$ the specific heat capacity at constant pressure and $R$ the gas constant for dry air.

From this set of equations we wish to construct a toy model that has large-scale geostrophically and hydrostatically balanced flow, permits intermittent convective-like behaviour and is of practical use for investigating issues that arise in the convective-scale DA problem (e.g. that it is cheap to integrate).

### 2.1 Modifications to the 3-D Euler equations

In order to derive a model with the properties outlined above Eqs. (1) are modified in two stages. Firstly, a set of physically based approximations are made and secondly a set of 'toy model' simplifications are made. The latter set does not attempt to replicate the real system, rather they are intended to retain desired physical characteristics of the real system but simplify the computational implementation. In order to simplify the system it will be assumed that the model is periodic in the zonal direction and homogeneous in the meridional direction (i.e. the variables are functions of longitude, height, and time only).

#### 2.1.1 Physically based modifications

The variables are decomposed such that they have a basic state and perturbation component as in e.g. Pielke (2001):

$$\Phi(x, z, t) = \Phi_0(z) + \Phi'(x, z, t). \tag{2}$$

Here $\Phi$ applies to any model variable except $\theta$ (for $\theta$ see below). The basic state (subscript 0) is a function of height only, and the perturbation (primed) is a function of longitude ($x$), height ($z$), and time ($t$). Potential temperature contains also a constant

reference value (subscript R):

$$\theta(x, z, t) = \theta_R + \theta_0(z) + \theta'(x, z, t). \tag{3}$$

The wind components $u$, $v$ and $w$ have zero reference state values, therefore the prime notation is dropped for the winds. For convenience, explicit reference to the arguments $x$ and $z$ will be dropped in much of the following derivation.

The basic state is assumed to satisfy hydrostatic balance:

$$\frac{\partial p_0}{\partial z} = -\rho_0 g, \tag{4}$$

and the equation of state is

$$p_0 = \rho_0 R \left( \frac{p_0}{p_{00}} \right)^\kappa (\theta_R + \theta_0). \tag{5}$$

The Brunt-Väisälä frequency, $N$, is defined as

$$N^2 = \frac{g}{\theta_R} \frac{d\theta_0}{dz}. \tag{6}$$

The pressure gradient terms in Eq. (1a) are represented with Eq. (2), products of perturbations are neglected, and it is assumed that $\rho_0 \gg \rho'$ in the momentum equations. A buoyancy variable, $b = b_0(z) + b'(x, z)$, is introduced for convenience, it is related to $\theta$ by

$$b = b_0(z) + b' = \frac{g}{\theta_R} (\theta_R + \theta_0(z) + \theta'). \tag{7}$$

Combining these physically based approximations gives the following equations:

$$\frac{\partial u}{\partial t} + \mathbf{u} \cdot \nabla u + \frac{1}{\rho_0} \frac{\partial p'}{\partial x} - fv = 0, \tag{8a}$$

$$\frac{\partial v}{\partial t} + \mathbf{u} \cdot \nabla v + fu = 0, \tag{8b}$$

$$\frac{\partial w}{\partial t} + \mathbf{u} \cdot \nabla w + \frac{1}{\rho_0} \frac{\partial p'}{\partial z} + \frac{g}{\rho_0} \rho' = 0, \tag{8c}$$

$$\frac{\partial \rho'}{\partial t} + \nabla \cdot (\rho \mathbf{u}) = 0, \tag{8d}$$

$$\frac{\partial b'}{\partial t} + \mathbf{u} \cdot \nabla b' + N^2 w = 0, \tag{8e}$$

$$p = \rho R \left( \frac{p}{p_{00}} \right)^\kappa \theta, \tag{8f}$$

$$b' = \frac{g}{\theta_R} \theta'. \tag{8g}$$

### 2.1.2   The "ABC model" modifications

It is desirable to reduce the stiffness of system Eqs. (8) so that it can be integrated explicitly with a time-step that is not too
small. The following 'toy model' modifications are made so that the toy equations retain the basic properties desired, i.e. be geostrophically and hydrostatically balanced on the large-scale but permit intermittent convection-like behaviour on the small-scale that is unbalanced. The modifications are as follows.

1. We control the gravity waves by replacing $N$ by the tunable parameter, $A$ (which has units of $\mathrm{s}^{-1}$). This is the pure gravity wave frequency (Sect. 4.4).

2. We control the acoustic waves by multiplying the divergent term of the compressible continuity equation by the dimensionless parameter $B$ (where $0 < B \leq 1$). To ensure energy conservation (Sect. 2.3) it is required that $B$ also multiplies the advective components of the momentum and thermodynamic equations. Acoustic waves can have frequencies that are normally much higher than gravity waves, but choosing a small $B$ can help to reduce the acoustic wave frequencies.

The effect of these parameters on the wave speeds will be demonstrated by numerical linear analysis in Sect. 4.6. The acoustic and gravity waves in the real atmosphere are coupled through the equation of state (Thuburn et al., 2002). This coupling can be reduced by using a linearised and simplified equation of state. Linearizing Eq. (8f) about the basic state gives

$$(1 - \kappa)p_0^{-\kappa}p' = \frac{\rho' R\theta_\mathrm{R}}{p_{00}^\kappa} + \frac{\rho_0 R\theta'}{p_{00}^\kappa}, \tag{9}$$

where we have used $\theta_\mathrm{R} + \theta_0 \approx \theta_\mathrm{R}$ (see Appendix A). This is used in two ways to give modifications 3 and 4 below.

3. Firstly, for the purpose of relating density and buoyancy perturbations in Eq. (8c), we neglect pressure perturbations in Eq. (9):

$$\frac{\rho'}{\rho_0} = -\frac{\theta'}{\theta_\mathrm{R}}, \tag{10}$$

which by Eq. (8g) equals $-b'/g$.

4. Secondly and separately, for the purposes of simplifying the equation of state, neglecting buoyancy perturbations in the linearised equation of state Eq. (9) gives

$$(1 - \kappa)p_0^{-\kappa}p' = \frac{\rho' R\theta_\mathrm{R}}{p_{00}^\kappa}. \tag{11}$$

This is a means of decoupling gravity and acoustic waves. Further, setting

$$C = \frac{R\theta_\mathrm{R}p_0^\kappa}{p_{00}^\kappa(1 - \kappa)}, \tag{12}$$

gives the simplified equation of state:

$$p' = C\rho', \tag{13}$$

where $C$ is taken to be a global constant, and has units of $\mathrm{Nmkg}^{-1} = \mathrm{m}^2\mathrm{s}^{-2}$. The quantity $\sqrt{BC}$ is the pure sound wave speed in this system (Sect. 4.5).

5. Reference density $\rho_0$ is taken to be a constant and not a function of height.

6. Define the scaled density perturbation, $\tilde{\rho}'$ as

$$\tilde{\rho}' = \frac{\rho'}{\rho_0}, \tag{14}$$

and with this definition, $\tilde{\rho}_0 = 1$.

Combining modifications 1 to 6 gives the final form of the toy model equations:

$$\frac{\partial u}{\partial t} + B\mathbf{u} \cdot \nabla u + C\frac{\partial \tilde{\rho}'}{\partial x} - fv = 0, \tag{15a}$$

$$\frac{\partial v}{\partial t} + B\mathbf{u} \cdot \nabla v + fu = 0, \tag{15b}$$

$$\frac{\partial w}{\partial t} + B\mathbf{u} \cdot \nabla w + C\frac{\partial \tilde{\rho}'}{\partial z} - b' = 0, \tag{15c}$$

$$\frac{\partial \tilde{\rho}'}{\partial t} + B\nabla \cdot (\tilde{\rho}\mathbf{u}) = 0, \tag{15d}$$

$$\frac{\partial b'}{\partial t} + B\mathbf{u} \cdot \nabla b' + A^2 w = 0. \tag{15e}$$

Note that Eq. (15d) conserves mass following the flow modulated by $B$, i.e. $B\mathbf{u}$, but total mass remains conserved (Sect. 2.3.1). We also include the following tracer transport equation for diagnostic purposes:

$$\frac{\partial q}{\partial t} + \mathbf{u} \cdot \nabla q = 0, \tag{16}$$

where $q$ is the tracer concentration. Note that the advection term is not multiplied by $B$ in Eq. (16) (as $B$ will be generally chosen as $B \le 1$, we allow advection of the tracer to have its full effect so that tracer transport can be seen over an integration of a few hours). We refer to these simplified equations as the "ABC model" reflecting the three tunable parameters.

## 2.2 Scale analysis of the "ABC model"

A scale analysis of Eqs. (15) is performed by non-dimensionalising the equations using characteristic values. Our scale analysis deviates from standard analyses in two ways: (i) we allow different characteristic length-scales for each variable (in the horizontal and vertical), and (ii) we do not assume incompressibility (see below for more explanation of this). For the characteristic values we set $u = \mathcal{U}u^*$, $v = \mathcal{V}v^*$, $w = \mathcal{W}w^*$, $\tilde{\rho}' = \mathcal{P}'\tilde{\rho}'^*$, $\tilde{\rho} \sim 1$, and $b' = \mathcal{B}'b'^*$. For the characteristic horizontal length-scales we set (respectively for each variable) $x = \mathcal{L}_u^H x_u^*$, $x = \mathcal{L}_v^H x_v^*$, $x = \mathcal{L}_w^H x_w^*$, $x = \mathcal{L}_{\tilde{\rho}'}^H x_{\tilde{\rho}'}^*$ and $x = \mathcal{L}_{b'}^H x_{b'}^*$, and for the vertical length-scales $z = \mathcal{L}_u^V z_u^*$, $z = \mathcal{L}_v^V z_v^*$, $z = \mathcal{L}_w^V z_w^*$, $z = \mathcal{L}_{\tilde{\rho}'}^V z_{\tilde{\rho}'}^*$ and $z = \mathcal{L}_{b'}^V z_{b'}^*$. The time-scale is set as $t = \left[\mathcal{L}_u^H/(B\mathcal{U})\right]t^*$. Upper case calligraphic variables (except $\mathcal{L}$) are characteristic values, starred variables are non-dimensional and $\mathcal{O}(1)$, and $\mathcal{L}_p^{H/V}$ represents the horizontal/vertical length-scale of variable $p$.

Often in scale analyses the characteristic vertical speed, $\mathcal{W}$, is written in terms of other characteristic variables by using the incompressible continuity equation in a 2-D (longitude-height) system $\partial u/\partial x + \partial w/\partial z = 0$. Scaling this gives $\mathcal{W} \sim \mathcal{U}\mathcal{L}_w^V/\mathcal{L}_u^H$. We do not use this relation as some of the flows considered are highly compressible.

Using these definitions in Eqs. (15) and introducing the Rossby number, $Ro = \mathcal{U}/f\mathcal{L}_u^H$, the aspect ratio, $\mathcal{A} = \mathcal{L}_u^V/\mathcal{L}_u^H$, the vertical-to-zonal wind ratio, $\mathcal{W}_{\mathcal{U}} = \mathcal{W}/\mathcal{U}$, and the meridional-to-zonal wind ratio, $\mathcal{V}_{\mathcal{U}} = \mathcal{V}/\mathcal{U}$ gives the following non-

dimensionalised equations:

$$BRo\left[\frac{\partial u^*}{\partial t^*}+u^*\frac{\partial u^*}{\partial x_u^*}+\mathcal{A}^{-1}\mathcal{W}_{\mathcal{U}}w^*\frac{\partial u^*}{\partial z_u^*}\right]+\frac{C\mathcal{P}'}{\mathcal{U}f\mathcal{L}_{\tilde{\rho}'}^{\mathrm{H}}}\frac{\partial\tilde{\rho}'^*}{\partial x_{\tilde{\rho}'}^*}-\mathcal{V}_{\mathcal{U}}v^* = 0,\tag{17a}$$

$$BRo\left[\frac{\partial v^*}{\partial t^*}+\frac{\mathcal{L}_u^{\mathrm{H}}}{\mathcal{L}_v^{\mathrm{H}}}u^*\frac{\partial v^*}{\partial x_v^*}+\frac{\mathcal{L}_u^{\mathrm{H}}}{\mathcal{L}_v^{\mathrm{V}}}\mathcal{W}_{\mathcal{U}}w^*\frac{\partial v^*}{\partial z_v^*}\right]+\mathcal{V}_{\mathcal{U}}^{-1}u^* = 0,\tag{17b}$$

$$BRo\left[\frac{\partial w^*}{\partial t^*}+\frac{\mathcal{L}_u^{\mathrm{H}}}{\mathcal{L}_w^{\mathrm{H}}}\frac{\partial w^*}{\partial x_w^*}+\frac{\mathcal{L}_u^{\mathrm{H}}}{\mathcal{L}_w^{\mathrm{V}}}\mathcal{W}_{\mathcal{U}}w^*\frac{\partial w^*}{\partial z_w^*}\right]+\frac{C\mathcal{P}'}{\mathcal{W}f\mathcal{L}_{\tilde{\rho}'}^{\mathrm{V}}}\frac{\partial\tilde{\rho}'^*}{\partial z_{\tilde{\rho}'}^*}-\frac{\mathcal{B}'}{\mathcal{W}f}b'^* = 0,\tag{17c}$$

$$\frac{\partial\tilde{\rho}'^*}{\partial t^*}+\frac{\partial\tilde{\rho}^*u^*}{\partial x_u^*}+\frac{\mathcal{L}_u^{\mathrm{H}}}{\mathcal{L}_w^{\mathrm{V}}}\mathcal{W}_{\mathcal{U}}\frac{\partial\tilde{\rho}^*w^*}{\partial z_w^*} = 0,\tag{17d}$$

$$BRo\left[\frac{\partial b'^*}{\partial t^*}+\frac{\mathcal{L}_u^{\mathrm{H}}}{\mathcal{L}_{b'}^{\mathrm{H}}}u^*\frac{\partial b'^*}{\partial x_{b'}^*}+\frac{\mathcal{L}_u^{\mathrm{H}}}{\mathcal{L}_{b'}^{\mathrm{V}}}\mathcal{W}_{\mathcal{U}}w^*\frac{\partial b'^*}{\partial z_{b'}^*}\right]+\frac{A^2\mathcal{W}}{\mathcal{B}'f}w^* = 0.\tag{17e}$$

When the first three terms of Eq. (17a) and Eq. (17b) are small (often achieved with small $Ro$) the geostrophic relationships emerge. Expressed back in terms of the dimensional variables they are

$$-fv+C\frac{\partial\tilde{\rho}'}{\partial x} = 0,\tag{18a}$$

$$u = 0.\tag{18b}$$

Under similar circumstances Eq. (17c) defines the hydrostatic relationship. Expressed back in terms of the dimensional variables it is

$$-b'+C\frac{\partial\tilde{\rho}'}{\partial z}=0.\tag{19}$$

## 2.3  Conservation of mass and energy

As the toy model equations (15) are no longer based on standard thermodynamics, we must demonstrate that they form a physically reasonable set. To this end we now show that they conserve mass and energy.

### 2.3.1  Conservation of mass

Noting definition (14), multiplying the continuity equation, Eq. (15d), by the constant $\rho_0$ gives the equation for the evolution of density perturbations. Adding the zero valued term $\partial\rho_0/\partial t$ then produces the equation for the evolution of density: $\partial\rho/\partial t+B\nabla\cdot(\rho\mathbf{u})=0$. Since the model uses periodic boundary conditions zonally, and zero vertical wind conditions at the top and bottom boundaries (Sect. 3.2), the divergence theorem shows that the equations conserve mass, $(\partial/\partial t)\left(\int\int dxdz\rho\right)=0$ (see Appendix B).

### 2.3.2  A useful 'identity' used to demonstrate conservation of energy

Dividing the continuity equation shown in Sect. 2.3.1 by $\rho_0$ gives the equation for $\tilde{\rho}$ evolution: $\partial\tilde{\rho}/\partial t+B\nabla\cdot(\tilde{\rho}\mathbf{u})=0$. Using this equation and expanding $\partial(\tilde{\rho}\gamma)/\partial t+B\nabla\cdot(\tilde{\rho}\gamma\mathbf{u})$, for an arbitrary time and space varying scalar field $\gamma$, we find:

$$\frac{\partial(\tilde{\rho}\gamma)}{\partial t}+B\nabla\cdot(\tilde{\rho}\gamma\mathbf{u})=\tilde{\rho}\left(\frac{\partial\gamma}{\partial t}+B\mathbf{u}\cdot\nabla\gamma\right).\tag{20}$$

Equation (20) is treated as an identity in the forthcoming energy analysis.

### 2.3.3 Kinetic energy

Multiplying respectively the momentum equations, (15a) to (15c), by $\tilde{\rho}u$, $\tilde{\rho}v$ and $\tilde{\rho}w$ and using (20) with $\gamma = u^2/2$, $\gamma = v^2/2$ and $\gamma = w^2/2$ we find:

$$\frac{\partial}{\partial t}\left(\frac{1}{2}\tilde{\rho}u^2\right) + B\nabla\cdot\left(\frac{1}{2}\tilde{\rho}u^2\mathbf{u}\right) + C\tilde{\rho}u\frac{\partial\tilde{\rho}'}{\partial x} - \tilde{\rho}ufv = 0, \tag{21a}$$

$$\frac{\partial}{\partial t}\left(\frac{1}{2}\tilde{\rho}v^2\right) + B\nabla\cdot\left(\frac{1}{2}\tilde{\rho}v^2\mathbf{u}\right) + \tilde{\rho}vfu = 0, \tag{21b}$$

$$\frac{\partial}{\partial t}\left(\frac{1}{2}\tilde{\rho}w^2\right) + B\nabla\cdot\left(\frac{1}{2}\tilde{\rho}w^2\mathbf{u}\right) + C\tilde{\rho}w\frac{\partial\tilde{\rho}'}{\partial z} - \tilde{\rho}wb' = 0. \tag{21c}$$

We can write the perturbation kinetic energy, $E_{\mathrm{k}}$, as

$$E_{\mathrm{k}} = \frac{\tilde{\rho}}{2}\left(u^2 + v^2 + w^2\right), \tag{22}$$

which allows the sum of Eq. (21a) to Eq. (21c) to be written

$$\frac{\partial}{\partial t}E_{\mathrm{k}} + B\nabla\cdot(E_{\mathrm{k}}\mathbf{u}) - \tilde{\rho}b'w + C\tilde{\rho}\mathbf{u}\cdot\nabla\tilde{\rho}' = 0. \tag{23}$$

### 2.3.4 Buoyant energy

Multiplying the thermodynamic equation (15e) by $\tilde{\rho}b'/A^2$ and using Eq. (20) with $\gamma = b'^2/(2A^2)$, we find

$$\frac{\partial}{\partial t}E_{\mathrm{b}} + B\nabla\cdot(E_{\mathrm{b}}\mathbf{u}) + \tilde{\rho}b'w = 0, \tag{24}$$

where the perturbation buoyant energy, $E_{\mathrm{b}}$, is

$$E_{\mathrm{b}} = \frac{\tilde{\rho}b'^2}{2A^2}. \tag{25}$$

### 2.3.5 Elastic energy

Multiplying the continuity equation (15d) by $C\tilde{\rho}'/B$ we find

$$\frac{\partial}{\partial t}E_{\mathrm{e}} + C\tilde{\rho}'\nabla\cdot(\tilde{\rho}\mathbf{u}) = 0, \tag{26}$$

where the perturbation elastic energy, $E_{\mathrm{e}}$, is

$$E_{\mathrm{e}} = \frac{C\tilde{\rho}'^2}{2B}. \tag{27}$$

### 2.3.6 Total combined energy and its conservation

Adding Eqs. (23), (24), and (26) shows that the combined energy, $E = E_k + E_b + E_e$, satisfies

$$\frac{\partial E}{\partial t} + B\nabla \cdot ((E_k + E_b)\mathbf{u}) + C\nabla \cdot (\tilde{\rho}'\tilde{\rho}\mathbf{u}) = 0. \tag{28}$$

Integrating Eq. (28) over the whole domain for the total combined energy gives

$$5 \quad \int \frac{\partial E}{\partial t} dV + \int B\nabla \cdot ((E_k + E_b)\mathbf{u})dV + C\int \nabla \cdot (\tilde{\rho}'\tilde{\rho}\mathbf{u})\, dV = 0. \tag{29}$$

This toy model is set-up to have periodic boundary conditions in the $x$-direction, to have no variation in the $y$-direction, and to have zero vertical wind at the top and bottom boundaries (see Section 3.2). The divergence theorem then leads to conservation of total combined energy (see Appendix B):

$$\frac{\partial}{\partial t}\left(\int E dV\right) = 0. \tag{30}$$

## 10  3  Numerical implementation of the "ABC model"

Now that a physically reasonable set of toy model equations has been formed, we now provide the details of how they are treated numerically.

### 3.1  Model discretization

The toy model uses a similar grid to that of the Southern UK (SUK) version of the UK Met Office's Unified Model (UM), but
with some differences given below. In the horizontal the SUK model covers a domain of $540\,\mathrm{km}$ in longitude and $432\,\mathrm{km}$ in latitude with a resolution of $1.5\,\mathrm{km}$ on an Arakawa-C grid. In the vertical it has 70 vertical levels up to approximately $40\,\mathrm{km}$ on an irregularly spaced Charney-Philips grid (Lean et al., 2008).

The toy model grid is shown in Fig. 1. The differences from the SUK are that the toy model is periodic in the zonal direction, is homogeneous in the meridional direction, and uses regularly spaced vertical levels up to a lid of about 15km. The toy model
uses only 60 levels (level spacing $\delta z \approx 250\,\mathrm{m}$) and has flat orography.

This grid is a natural discretization of the equations which does not require a significant number of interpolations. There are approximately $10^5$ variables in the state space of the toy system.

### 3.2  Boundary conditions

The vertical boundary conditions that we use are summarized in Table 2. At the lower boundary the horizontal winds are zero
(no-slip conditions) and the vertical wind is zero to conserve total mass and energy. At the upper and lower boundaries the vertical derivative of density is zero. For the equations to have the capability to support hydrostatic balance, Eq. (19) implies that $b'$ should be zero at the vertical boundaries. At the upper boundary the horizontal winds are chosen to maintain consistency with the boundary conditions of $\tilde{p}'$ and $b'$ through thermal wind balance and the vertical wind is again zero to conserve total mass and energy.

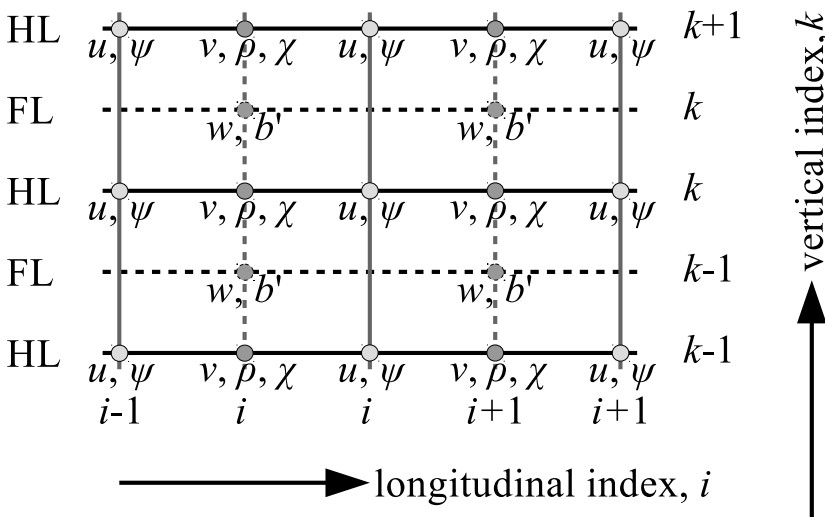

**Figure 1.** The arrangement of variables on the toy model's grid: an Arakawa-C grid in the horizontal and a Charney-Phillips grid in the vertical. Note the abbreviations: FL=Full Level and HL=Half Level.

## 3.3 Numerical differentiation and integration

### 3.3.1 Time integration scheme

The time integration is evaluated using a split explicit, forward-backward scheme (Cullen and Davies, 1991) and here we give a description of this scheme applied to Eqs. (15). The forward-backward scheme operates over a time-step $\Delta t$ and comprises two stages: an adjustment stage and an advection stage.

**Adjustment stage**

The adjustment stage operates over a sub-timestep $\delta t$, where $\delta t = \Delta t/n$ and $n$ is typically a small positive integer (in this implementation $n = 2$). The adjustment stage contains two parts: the forward part and the backward part. Let $t$ be the time at the start of the $\Delta t$ timestep and let $t_i$ be shorthand for $t + i\delta t$. The following is a description of the $i$th sub-timestep.

In the forward part of the forward-backward scheme, the momentum and thermodynamic equations are evaluated omitting the advective terms. The $u$ and $v$ equations are considered simultaneously to find the adjustment due to the Coriolis and pressure gradient terms. Then the $w$-momentum and $b'$ equations are dealt with simultaneously to find the adjustment due to buoyancy, pressure gradient and vertical wind. The forward part of the adjustment stage gives an implicit approximation to $u$, $v$, $w$ and $b'$ at the next sub-timestep.

The equations for $u$ Eq. (15a) and $v$ Eq. (15b), omitting the advective terms are discretized as

$$u(t_{i+1}) = u(t_i) - \delta t C \frac{\partial \tilde{\rho}'(t_i)}{\partial x} + \frac{\delta t f}{2}\left(v(t_i) + v(t_{i+1})\right), \tag{31a}$$

$$v(t_{i+1}) = v(t_i) - \frac{\delta t f}{2}\left(u(t_i) + u(t_{i+1})\right). \tag{31b}$$

Solving Eqs. (31) for $u(t_{i+1})$ and $v(t_{i+1})$ gives

$$5 \quad u(t_{i+1}) = \frac{\beta_f}{\alpha_f} u(t_i) - \frac{\delta t C}{\alpha_f}\frac{\partial \tilde{\rho}'(t_i)}{\partial x} + \frac{\delta t f}{\alpha_f} v(t_i), \tag{32a}$$

$$v(t_{i+1}) = \frac{\beta_f}{\alpha_f} v(t_i) - \frac{\delta t f}{\alpha_f} u(t_i) + \frac{\delta t^2 C f}{2\alpha_f}\frac{\partial \tilde{\rho}'(t_i)}{\partial x}, \tag{32b}$$

where $\alpha_f$ and $\beta_f$ are defined by

$$\alpha_f = 1 + \frac{\delta t^2 f^2}{4}, \quad \text{and} \quad \beta_f = 1 - \frac{\delta t^2 f^2}{4}. \tag{33}$$

The equations for $w$ Eq. (15c) and $b'$ Eq. (15e) omitting advective terms are discretized as

$$10 \quad w(t_{i+1}) = w(t_i) - \delta t C \frac{\partial \tilde{\rho}'(t_i)}{\partial z} + \frac{\delta t}{2}\left(b'(t_i) + b'(t_{i+1})\right), \tag{34a}$$

$$b'(t_{i+1}) = b'(t_i) - \frac{\delta t A^2}{2}\left(w(t_i) + w(t_{i+1})\right). \tag{34b}$$

Solving Eqs. (34) for $w(t_{i+1})$ and $b'(t_{i+1})$ gives

$$w(t_{i+1}) = \frac{\beta_A}{\alpha_A} w(t_i) - \frac{\delta t C}{\alpha_A}\frac{\partial \tilde{\rho}'(t_i)}{\partial z} + \frac{\delta t}{\alpha_A} b'(t_i), \tag{35a}$$

$$b'(t_{i+1}) = \frac{\beta_A}{\alpha_A} b'(t_i) - \frac{\delta t A^2}{\alpha_A} w(t_i) + \frac{\delta t^2 C A^2}{2\alpha_A}\frac{\partial \tilde{\rho}'(t_i)}{\partial z}, \tag{35b}$$

where $\alpha_A$ and $\beta_A$ are defined by

$$\alpha_A = 1 + \frac{\delta t^2 A^2}{4}, \quad \text{and} \quad \beta_A = 1 - \frac{\delta t^2 A^2}{4}. \tag{36}$$

Equations (32a), (32b), (35a) and (35b) are the discretized forms of the split-explicit equations that are evaluated in the forward part of the forward-backward scheme in the adjustment stage. The spatial derivatives are left in continuous form, but are discretized in the numerics using standard centred finite-differences.

In the backward part of the forward-backward scheme the continuity equation (15d) is evaluated using the wind and buoyancy data calculated in the forward part, i.e.

$$\tilde{\rho}'(t_{i+1}) = \tilde{\rho}'(t_i) - \delta t B \left(\tilde{\rho}(t_i)\nabla \cdot \mathbf{u}(t_{i+1}) + \mathbf{u}(t_{i+1}) \cdot \nabla \tilde{\rho}(t_i)\right). \tag{37}$$

The term in brackets on the right hand side is equal to $\nabla \cdot (\tilde{\rho}(t_i)\mathbf{u}(t_{i+1}))$, but has been expanded in Eq. (37) to allow the second term to use the upstream gradient of $\tilde{\rho}(t_i)$. After integration of $n$ steps over the full $\Delta t$ the value of $\tilde{p}'$ is known and the values

of the variables $u$, $v$, $w$ and $b'$ are known but without the effect of advection.

**Advection stage**

The advection stage advects the fields $u$, $v$, $w$ and $b'$ calculated in the adjustment stage using the sub-timestep-averaged winds $\bar{u}$ and $\bar{w}$, which are taken to be valid over the full $\Delta t$. Let $\phi$ be any of $u$, $v$, $w$ or $b'$, then the advection step is given by

$$\phi(t + \Delta t) = \phi(t) - \Delta t B \bar{\mathbf{u}} \cdot \nabla \phi(t), \tag{38}$$

where $\bar{\mathbf{u}} = (\bar{u}, \bar{w})^{\mathrm{T}}$. As with Eq. (37), the upwind gradient of $\phi$ is computed in Eq. (38). Note that for the tracer advection, $\phi = q$, Eq. (38) is used with $B = 1$.

**Overall properties**

The spatial derivatives evaluated by the forward-upstream scheme are first order accurate (Press et al., 2007) and the time integration which utilises a split-explicit and forward-backward scheme is also first-order accurate (Ames, 1958; Gadd, 1978).

The stability of the forward-backward scheme increases the time-steps which are permitted by the CFL criterion (Ames, 1958). The split-explicit scheme has been used in early implementations of the UK Met Office's NWP model due to its ability to conserve mass (Gadd, 1978; Cullen and Davies, 1991).

## 4   Linear/normal mode analysis of the "ABC model"

In this section a normal mode analysis of the toy model equations is performed. This follows a similar procedure used for

the shallow water equations in Section 6.4 of Daley (1991) and in Section 2.4 of Cullen (2006). In this procedure the model equations are first linearised to permit a mathematical analysis of small amplitude perturbations. The normal modes are the characteristic solutions of the linear equations. Each mode has a characteristic frequency and wavelength (hence the analysis is done in terms of spectral modes), and is a specific combination of different variables. There are three different types of normal mode solution, namely Rossby-like modes (whose normal mode patterns obey geostrophic and hydrostatic balances),

gravity modes (buoyancy driven modes which have characteristic horizontal divergence), and acoustic modes (the most rapidly oscillating modes which are made up of compression waves). The linear/normal mode analysis allows us to probe the dispersion relations (mode frequencies as a function of wavenumber) and the balanced/unbalanced character of the linear modes. For simplicity this analysis is performed on a continuous periodic domain of size $L_x$ and $L_z$.

### 4.1   Linearisation

The non-linear model equations (15) are linearised about a state of rest (see Appendix A). It is convenient to write the model equations in terms of velocity potential, $\chi$ (describing the divergent part of the horizontal flow), and streamfunction, $\psi$ (describing the rotational or solenoidal part). The Helmholtz theorem (see e.g. Salby, 1996, which is generically given by "horizontal wind $= \nabla_{\mathrm{h}} \chi + \mathbf{k} \times \nabla_{\mathrm{h}} \psi$", where $\mathbf{k}$ is the vertical unit vector, and $\nabla_{\mathrm{h}} = (\partial/\partial x, \partial/\partial y, 0)$) allows the horizontal

wind to be written as

$$
\begin{pmatrix} u \\ v \\ 0 \end{pmatrix} = \begin{pmatrix} \partial\chi/\partial x \\ \partial\chi/\partial y \\ 0 \end{pmatrix} + \begin{pmatrix} 0 \\ 0 \\ 1 \end{pmatrix} \times \begin{pmatrix} \partial\psi/\partial x \\ \partial\psi/\partial y \\ 0 \end{pmatrix} = \begin{pmatrix} \partial\chi/\partial x \\ \partial\chi/\partial y \\ 0 \end{pmatrix} + \begin{pmatrix} -\partial\psi/\partial y \\ \partial\psi/\partial x \\ 0 \end{pmatrix}.
\tag{39}
$$

Noting the lack of $y$-dependence in the ABC model, this gives: $u = \partial\chi/\partial x$ and $v = \partial\psi/\partial x$. The linearised model equations are then:

$$
\frac{\partial}{\partial t}\frac{\partial^2\chi}{\partial x^2} + C\frac{\partial^2\tilde{\rho}'}{\partial x^2} - f\frac{\partial^2\psi}{\partial x^2} = 0,
\tag{40a}
$$

$$
\frac{\partial}{\partial t}\frac{\partial^2\psi}{\partial x^2} + f\frac{\partial^2\chi}{\partial x^2} = 0,
\tag{40b}
$$

$$
\frac{\partial w}{\partial t} + C\frac{\partial\tilde{\rho}'}{\partial z} - b' = 0,
\tag{40c}
$$

$$
\frac{\partial\tilde{\rho}'}{\partial t} + B\frac{\partial^2\chi}{\partial x^2} + B\frac{\partial w}{\partial z} = 0,
\tag{40d}
$$

$$
\frac{\partial b'}{\partial t} + A^2 w = 0,
\tag{40e}
$$

## 4.2 Normal mode analysis

Now take the following functional dependence for a particular frequency $\sigma$, horizontal wavenumber $k$ and vertical wavenumber $m$:

$$
\begin{vmatrix} \chi(x,z,t) \\ \psi(x,z,t) \\ w(x,z,t) \\ \tilde{\rho}'(x,z,t) \\ b'(x,z,t) \end{vmatrix} = \begin{vmatrix} i & \hat{\chi} \\ 1 & \hat{\psi} \\ k/L_x & \hat{w} \\ k/L_x\sqrt{B/C} & \hat{\tilde{\rho}}' \\ -iAk/L_x & \hat{b}' \end{vmatrix} \exp\left[ i\left\{ \left(\frac{kx}{L_x}\right) + \left(\frac{mz}{L_z}\right) - \sigma t \right\} \right].
\tag{41}
$$

Substituting Eq. (41) into Eq. (40) and expressing the resulting set of equations in matrix form gives

$$
(\mathbf{L} - \sigma\mathbf{I}) \begin{vmatrix} \hat{\chi} \\ \hat{\psi} \\ \hat{w} \\ \hat{\tilde{\rho}}' \\ \hat{b}' \end{vmatrix} = 0,
\tag{42}
$$

where

$$
\mathbf{L} = \begin{pmatrix} 0 & f & 0 & -\frac{k\sqrt{BC}}{L_x} & 0 \\ f & 0 & 0 & 0 & 0 \\ 0 & 0 & 0 & \frac{m\sqrt{BC}}{L_z} & A \\ -\frac{k\sqrt{BC}}{L_x} & 0 & \frac{m\sqrt{BC}}{L_z} & 0 & 0 \\ 0 & 0 & A & 0 & 0 \end{pmatrix}.
\tag{43}
$$

This is an eigenvalue equation where $\mathbf{L}$ is a real and symmetric matrix (due to the choice of factors in Eq. (41)), and so will have real eigenvalues. For each distinct choice of horizontal and vertical wavenumber $(k, m)$, $\mathbf{L}$ has five eigenvalues, denoted $\sigma_{\mathrm{R}}$, $\sigma_{\mathrm{g}}$, $\sigma_{\mathrm{g}'}$, $\sigma_{\mathrm{a}}$, and $\sigma_{\mathrm{a}'}$ where

$$\sigma_{\mathrm{R}} = 0, \quad \sigma_{\mathrm{g}} = -\sigma_{\mathrm{g}'}, \quad \text{and} \quad \sigma_{\mathrm{a}} = -\sigma_{\mathrm{a}'}. \tag{44}$$

The three distinct modes are the Rossby-like mode (subscript "R"), two inertia gravity modes ("g" and "g'"), and two acoustic modes ("a" and "a'"). The algebraic form of the R mode is simple and is discussed in Sect. 4.3 below, but the forms of the remaining modes are very complicated and so are considered only firstly in 'pure' forms (Sects. 4.4 and 4.5) and then numerically in the wave speed analysis (Sect. 4.6).

## 4.3 The Rossby-like mode

The normalized R mode has $\sigma_{\mathrm{R}} = 0$ and is

$$\begin{pmatrix} \hat{\chi} \\ \hat{\psi} \\ \hat{w} \\ \hat{\rho}' \\ \hat{b}' \end{pmatrix} = \frac{1}{\sqrt{K}} \begin{pmatrix} 0 \\ -\frac{A}{f} \frac{k}{m} \frac{L_z}{L_x} \\ 0 \\ -\frac{A L_z}{m \sqrt{BC}} \\ 1 \end{pmatrix}, \tag{45}$$

where $K = (A L_z [k B C + L_x f^2] + L_x^2 f^2 m^2 C)/(L_x^2 f^2 m^2 BC)$. This mode, as we shall show, supports geostrophic balance defined by Eqs. (18a) and (18b). Firstly, relation (18a) in terms of the variables defined in Eq. (41) and for wavenumber $k$ is

$$\frac{k}{L_x} \sqrt{BC} \hat{\rho}' = f \hat{\psi}, \tag{46}$$

which is consistent with Eq. (45). Secondly, and trivially, relation (18b) is equivalent to $\partial \chi / \partial x = 0$, which is also consistent with Eq. (45). There is no vertical wind associated with the R mode. There remains a buoyancy component for this mode to support hydrostatic balance defined by Eq. (19). Relation (19) in terms of the variables defined in Eq. (41) and for wavenumbers $k, m$ is

$$\frac{m \sqrt{BC}}{L_z} \hat{\rho}' = -A \hat{b}', \tag{47}$$

which is consistent with Eq. (45).

## 4.4 The pure gravity waves

Following Kalnay (2002) pure gravity waves can be investigated by neglecting rotation and pressure perturbations (by Eq. (13) density perturbations are therefore neglected too)). We anticipate that the gravity waves will be sensitive to $A$ given that $A$ is related to the static stability parameter $N$ (the Brunt-Väisälä frequency). Under these conditions, Eq. (43) has two eigenvalues, 25 $\sigma_{\mathrm{g}} = A$ and $\sigma_{\mathrm{g}'} = -A$, representing the pure gravity wave frequencies. In the limit of $A = 0$, no gravity waves are supported.

## 4.5 The pure acoustic waves

Following Kalnay (2002) pure acoustic waves can be investigated by neglecting rotation, gravitation, and stratification (i.e. set $f = 0$, $g = 0$, $A^2 = 0$, and $b' = 0$). Under these conditions, Eq. (43) has three eigenvalues, $\sigma = 0$ (which is an incompressible mode that does not interest us here), $\sigma_a = \sqrt{BC}\sqrt{(k/L_x)^2 + (m/L_z)^2}$ and $\sigma_{a'} = -\sigma_a$, the latter two representing the pure acoustic wave frequencies. The pure acoustic wave speed in the horizontal (e.g.) is $\partial \sigma_a / \partial k$, which becomes $\sqrt{BC}$ in the small-scale limit. In the limit that $B = 0$ or $C = 0$, the system becomes incompressible and no acoustic waves are supported.

## 4.6 Wave speed analysis experiments

In sections 4.4 and 4.5, we demonstrated how the pure gravity and acoustic waves depend upon the parameters $A$, $B$ and $C$. The analysis there was simplified (by explicitly neglecting processes that are not directly associated with gravity and acoustic waves respectively) in order to derive analytical expressions. Here we look at the gravity and acoustic wave speeds in a more detailed way without making the approximations made before. These reveal the normal modes of the linearised system Eqs. (40) (see e.g. Thuburn et al. (2002)), which now include rotation, gravitation, and stratification. We show how the wave speeds behave in the linearised system, and as a function of wavenumber, and of parameter values. To reduce the stiffness of the system we would like the speeds of the gravity and acoustic modes to have value $\sim \mathcal{U}$ or $\sim \mathcal{V}$, the characteristic speeds of the horizontal wind components, and so the results of this subsection are important for choosing parameter values for suitable model runs.

The standard values of the parameters that we use for this section are: $A = 0.02\,\mathrm{s}^{-1}$ (estimated from a typical value of the Brunt-Väisälä frequency), $B = 1.0$, $C = 10^5\,\mathrm{m}^2\mathrm{s}^{-2}$ (estimated from initialising data), and $f = 10^{-4}\,\mathrm{s}^{-1}$, and for simplicity, periodicity is assumed in the $x$ and $z$ directions.

Figure 2 shows the horizontal group speeds for the gravity ($c_g = \partial \sigma_g / \partial k$, panel a) and acoustic ($c_a = \partial \sigma_a / \partial k$, panel b) waves as a function of the integer index, $n_x$ (characterising the horizontal wavenumber $k = 2n_x\pi/L_x$) for a range of parameter values (the integer index, $n_z$, characterising the vertical wavenumber $m = 2n_z\pi/L_z$, is fixed at $n_z = 3$). Note that these $k$ and $m$ are slightly different from those used in Sects. 4.1 to 4.5.

Gravity waves in the approximated system are found to be stationary (Sect. 4.4), but gravity waves in the full system are not, see Fig. 2a. There is a strong sensitivity of $c_g$ to $A$ (larger $A$, faster gravity waves), and the fastest gravity waves have large horizontal, and large vertical scales (small $n_x$ and $n_z$). There is also a sensitivity of $c_g$ to $BC$, which is especially evident at large vertical scales and over large and intermediate horizontal scales (not shown, but see Sect. 5.3.3).

Acoustic waves in this system have different characteristics to the gravity waves in many respects. Acoustic waves can be much faster, but their speed may be controlled via the strong sensitivity of $c_a$ to $BC$, and the fastest acoustic waves have small horizontal, and small vertical scales (large $n_x$ and $n_z$), see Fig 2b. It is at these small scales that the acoustic waves saturate to the value $\sqrt{BC}$ as found in Sect. 4.5. The sensitivity of $c_a$ to $A$ is weak for the smaller values of $A$ tested, but moderate for the largest value of $A$ tested (not shown).

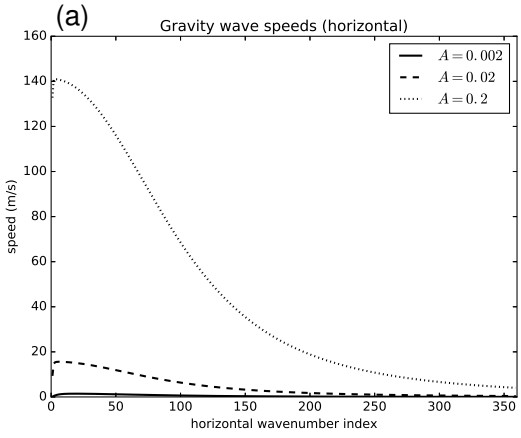
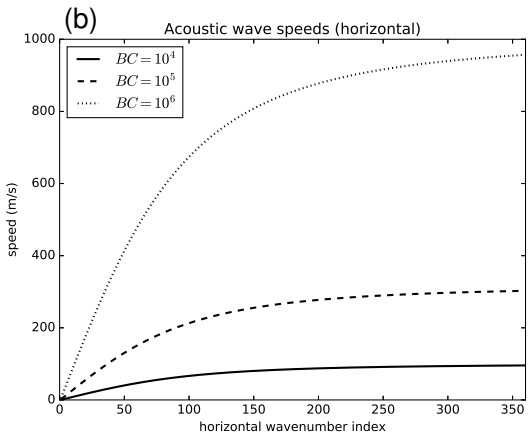

**Figure 2.** Panel (a) sensitivity of the horizontal gravity wave group speed to the tunable parameter $A$ (in $\text{s}^{-1}$), where $BC = 10^5\,\text{m}^2\text{s}^{-2}$. Panel (b) sensitivity of the horizontal acoustic wave group speed to $BC$ (in $\text{m}^2\text{s}^{-2}$), where $A = 0.02\,\text{s}^{-1}$. In both panels $f = 10^{-4}\,\text{s}^{-1}$ and the vertical wavenumber index is $n_z = 3$.

The ability of the parameters $A$, $B$ and $C$ to change the speed of both the horizontal gravity and acoustic waves has been demonstrated in Fig. 2. The buoyancy frequency parameter $A$, primarily controls the gravity waves, and the product $BC$ primarily controls the acoustic waves. The vertical gravity and acoustic wave speeds respond in a similar way to the parameters as the horizontal waves (not shown). In addition to modifying the acoustic wave speeds, $B$ and $C$ have other, separate effects –

$B$ slows advection round the domain, and $C$ influences the hydrostatic and geostrophic balance relationships (see section 2.2). It is permissible to alter the value of only one or any combination of these parameters depending on the required result. These results are used in the next section to help choose suitable parameter values.

### 4.7 Reference parameters

In this section some desired dynamical characteristics that are seen in the real atmosphere are demonstrated in this simplified

setup. It is required that the model mimics the multi-scale behaviour of the real atmosphere, i.e. displays hydrostatic and geostrophic balance on the large-scale while permitting imbalance and intermittent convective-like behaviour on the small-scale, while allowing an explicit solver. The results from the linear analysis of Sect. 4.6 gives a taste of how the wave speeds depend on $A$, $B$ and $C$, but the values that we settle on as reference values are $A = 2\times10^{-2}\text{s}^{-1}$, $B = 10^{-2}$ and $C = 10^4\text{m}^2\text{s}^{-2}$.

Figure 3a shows the frequencies, and the magnitudes of the horizontal and vertical wave speeds for the gravity and acoustic

waves for these reference parameters. The acoustic wave frequencies in panel a are always higher than those of the gravity waves (the latter, have an upper bound of $A$). The frequencies of the gravity and acoustic waves for $n_z = 3$ (left) are of the same order, but the acoustic wave frequencies for the extreme case of $n_z = 59$ (right), have much higher values by more than an order of magnitude. These are classic dispersion curves for these modes in the atmosphere (e.g. Fig. 14.9 of Salby (1996)) and they allow us to estimate that the highest frequency that the model will encounter with these parameters is $\sim 0.25\,\text{s}^{-1}$ (4 s

period, from the right panel of Fig. 3a). The parameter values though will (in Sect. 5.3) be changed by an order of magnitude each. Over these extended parameter values the maximum wave frequency is found to be $\sim 1.6\,\mathrm{s}^{-1}$ (0.625 s period). This allows us to set the time steps of our model (Sect. 3.3.1), which we choose as $\Delta t = 0.1\,\mathrm{s}$ and $\delta t = 0.05\,\mathrm{s}$. We use these values for all experiments.

The ability to control speeds in order to make the gravity and acoustic wave speeds comparable is more effective than the ability to control frequencies. Comparing, for instance, Figs. 3b and 2 shows how the gravity and acoustic wave speeds have been reduced to comparable values (a maximum of $10\,\mathrm{ms}^{-1}$ with the reference parameters, compared with a maximum of $1000\,\mathrm{ms}^{-1}$ for the parameters tested for Fig. 2). The speed of $10\,\mathrm{ms}^{-1}$ applies in the horizontal (Fig. 3b) and in the vertical (Fig. 3c). As well as using a $\Delta t$ that is smaller than the shortest wave period (see above), we can use the maximum

wave speed to also check that $\Delta t$ is consistent with the grid spacings ($\Delta x$ and $\Delta z$) via the Courant-Fredricks-Lewy (CFL) condition (Durran, 1999). Typically this states that the Courant number, $Co$, satisfies $0 \leq Co \leq 1$ for stability of the numerical solution. Given that $\Delta x = 1500\,\mathrm{m}$, $\Delta z = 250\,\mathrm{m}$, and $\Delta t = 0.1\,\mathrm{s}$, the Courant number is $Co = \Delta t \times (u_{\mathrm{max}}/\Delta x + u_{\mathrm{max}}/\Delta z) = 0.1 \times (10/1500 + 10/250) \approx 0.005$, which satisfies the CFL condition for the reference parameters. The maximum wave speed for the other parameter sets studied in Sect. 5.3 is $\sim 30\,\mathrm{ms}^{-1}$, which still results in a small Courant number.

# 5    "ABC model" integration results

## 5.1    Idealised initial conditions

The model was first initialised with idealised initial conditions to ensure that the model behaves reasonably with the reference parameter values. In this run the initial conditions are zero for all variables apart from $\tilde{\rho}'$, which takes the form of the Gaussian described in the caption (panel a). The $\tilde{\rho}'$ and $(u, v)$ fields for up to six hours are shown in Fig. 4. In the real atmosphere such

a positive density perturbation induces anticyclonic motion as geostrophic adjustment develops, and a similar response is seen in the toy model (the 'vertical' components of the arrows represent meridional wind, which is out of the page on the right and into the page on the left). After three hours (panel b), the horizontal wind is significantly divergent indicating that the $\tilde{\rho}'$ perturbation is being dissipated by gravity waves which act smooth-out the initial perturbation, whose maximum value has reduced to about a third of its original value. After six hours (panel c) the flow is mainly rotational (there is a weak convergent

flow near the centre of the domain) and the $\tilde{\rho}'$ perturbation has moved to the boundaries.

Figure 4 can be used to verify the wave speeds determined by linear analysis. Consider Fig. 4b, where the edge of the feature has propagated approximately $80\,\mathrm{km}$ over the three hours. This gives an approximate horizontal gravity wave speed of $\sim 7\,\mathrm{ms}^{-1}$, which is around the maximum horizontal gravity wave speed found from the linear analysis in Fig. 3b.

## 5.2    Intermittent convection-like behaviour

Convective motion in the atmosphere is difficult to model and to assimilate as it is often intermittent and associated with small-scale divergence. In the real atmosphere it is usually driven by latent heating, but our simple model is dry and so we rely on

**Figure 3.** Gravity and acoustic wave properties for the reference parameters $A = 2 \times 10^{-2}\text{s}^{-1}$, $B = 10^{-2}$ and $C = 10^4\text{m}^2\text{s}^{-2}$. The panels are: frequencies (a), and the magnitudes of the horizontal (b) and vertical (c) wave speeds. In (a) and (b) values are a function of horizontal wavenumber, $n_x$, and the left column is for $n_z = 3$, and the right column is for $n_z = 59$. In (c) values are a function of vertical wavenumber, $n_z$, and the left column is for $n_x = 10$, and the right column is for $n_x = 350$.

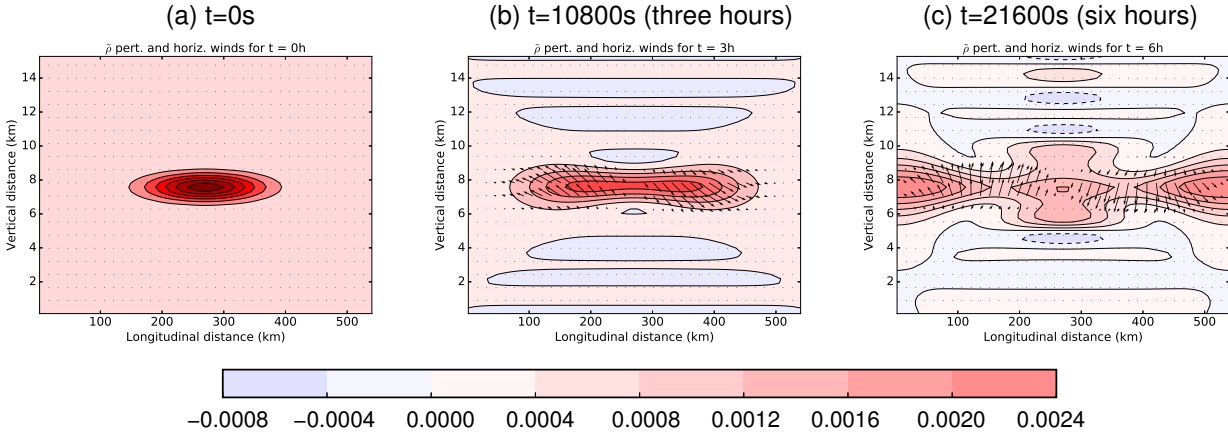

**Figure 4.** Model integration of the density perturbation $\tilde{\rho}'$ (colours), and horizontal wind vectors. The initial conditions are zero for all variables apart from $\tilde{\rho}'$, which takes the form of a Gaussian with an amplitude 0.01, a horizontal length-scale of 90 km, and a vertical length-scale of 700 m. The Gaussian is positioned in the middle of the domain. Parameters have the reference values $A = 2 \times 10^{-2}\,\mathrm{s}^{-1}$, $B = 10^{-2}$ and $C = 10^4\,\mathrm{m}^2\mathrm{s}^{-2}$. Note that the $y$-component of each wind arrow is the meridional, not the vertical, component of the wind. At six hours the maximum magnitude of the $u$-wind is $\sim 1.4\,\mathrm{m\,s}^{-1}$ and the maximum value of the $v$-wind is $\sim 3.6\,\mathrm{m\,s}^{-1}$.

other processes such as wave breaking to drive such motion. Intermittent convection-like motion is a desirable property of our model in order for it to have a significant unbalanced component on the small-scale, and hence be a useful system to study convective-scale data assimilation.

An indication of the presence of convection is vertical motion, and vertical motion necessarily indicates an imbalance – see
5   Eq. (45). We look at the $w$ field from an integration of the model firstly with the reference parameters. The initial conditions of the model were created from the following procedure.

– Take values of $u$, $v$ from a latitude/height slice of an output the Met Office's convective-scale (1.5 km grid) UM (this is the same model used by the Met Office during the 2012 Olympics (Golding et al., 2014), and has the same horizontal resolution and grid staggering as our model). These fields are adjusted to eliminate the discontinuity imposed by the
10   periodic boundary conditions[1].

– Calculate $\tilde{\rho}'$ by integrating the geostrophic balance equation (18a) on each level.

– Calculate $b'$ from the hydrostatic balance equation (19) for each horizontal location.

– Calculate $w$ from the continuity equation for zero three-dimensional divergence.

---

[1]This is done by incrementing the left half of the domain by the amount $-((\Delta - \delta)/2)\exp\left[-(x/\ell)^2\right]$, and the right half by $+((\Delta - \delta)/2)\exp\left[-((x - L_x)/\ell)^2\right]$, where $x$ is the horizontal distance from the western boundary, $\ell = 150$km is the relaxation distance, $\Delta$ is the size of the discontinuity in $u$ or $v$ (i.e. the magnitude of the difference in $u$ or $v$ between the western and eastern boundaries in the raw UM data), and $\delta$ is the magnitude of a typical increment of $u$ or $v$ between neighbouring grid-boxes. This procedure is performed separately for each vertical level.

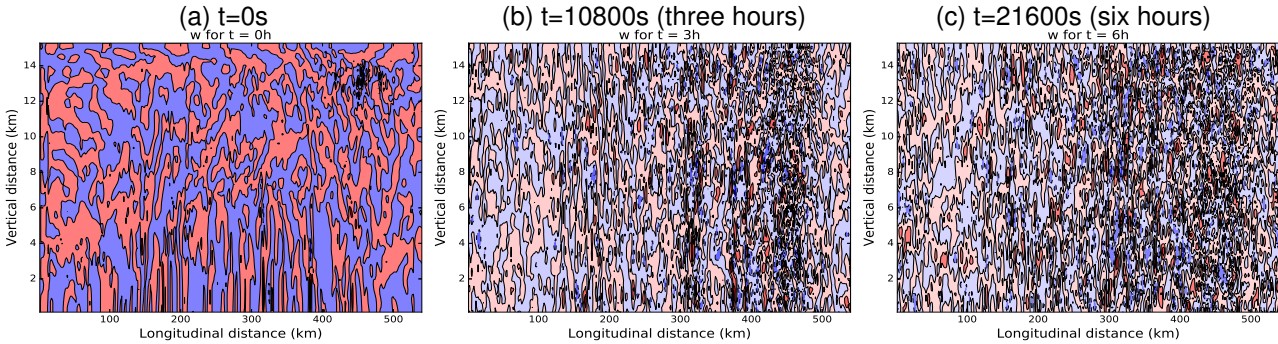

**Figure 5.** Model integration of the vertical wind $w$ up to six hours. The initial conditions in panel a are derived from an output of the UM as described in the text (Sect. 5.2). Parameters have the reference values $A = 2 \times 10^{-2}\,\mathrm{s}^{-1}$, $B = 10^{-2}$ and $C = 10^4\,\mathrm{m}^2\mathrm{s}^{-2}$. At the initial time the maximum magnitude of $w$ is $\sim 0.6\,\mathrm{ms}^{-1}$ and at six hours it is $\sim 0.16\,\mathrm{ms}^{-1}$. Red (blue) indicates positive (negative) values of $w$.

Each variable is then incremented (independently for each level) so that its horizontal mean is zero, and finally $\tilde{\rho}$ is set as $\tilde{\rho} = 1 + \tilde{\rho}'$. The model's initial conditions are then nearly balanced (the incrementing will disrupt the hydrostatic balance slightly), and unbalanced motion (including convection-like behaviour) then develops.

   Figure 5 shows $w$ over a six hour integration of the model using the reference parameters. The initial conditions in panel a show vertical winds that are of relatively small-scale in the horizontal, with elongated structures over the lowest $5\,\mathrm{km}$ or so in the middle of the domain. These are of course not generated by this model, but are derived from the UM data. An indication of the kind of behaviour that this model is capable of generating are shown in panels b and c, for three and six hours into the forecast respectively. The most striking aspect of the $w$ field at three and six hours is that the scales of the features are even smaller than those at $t = 0$. Additionally the magnitudes of $w$ are smaller with very small regions of moderate values especially in the eastern part of the domain. The similarity in these qualitative aspects of panels b and c shows that this kind of behaviour is not merely transient. We regard these plots as indicators of intermittent convection-like behaviour, which is studied further below.

### 5.3 Systematic exploration of model behaviour over parameter space

In order to test novel approaches to data assimilation it is desirable to run the model in different flow regimes, which we investigate by varying the parameters $A$, $B$, and $C$ systematically, each over a three-hour model run. We are particularly interested in understanding how the parameter values affect the degree of convection and of imbalance, and we start this investigation by introducing the diagnostics for the reference parameters.

#### 5.3.1 Reference parameters

We settle on four kinds of diagnostic for each parameter set, which are shown in Fig. 6 for the reference parameters. Panel a is the vertical wind speed, and panel b is the effective buoyancy, $bu_{\mathrm{eff}}$. The latter is defined as the (non-constant) stability $bu_{\mathrm{eff}} =$

$\partial \left( b_0(z) + b'(x,z) \right) / \partial z$, where $\partial b_0(z)/\partial z = A^2$. A positive (negative) effective buoyancy indicates statically (un)stable air, so negative and small positive values suggest convective activity (contours for negative values of $w$ and $bu_{\text{eff}}$ are dashed). Panel c is the distribution of tracers after three hours. The tracers were initialised at $t = 0$ on a grid of 20 points distributed throughout the domain (small rectangular regions in panel c) and the distribution after three hours provides an indication of the history of the wind behaviour. These fields are labeled with the minimum, maximum, and root-mean-squared values. Panel d indicates the degree of relative geostrophic imbalance (blue lines and left scale) and hydrostatic imbalance (red lines and right scale) averaged over the domain, at half-hour intervals over the integration. These quantities are found respectively using Eq. (18a) and Eq. (19) to give

$$\text{geo. imbal} \quad = \quad \text{rms}\left[ \left( C\frac{\partial \tilde{\rho}'}{\partial x} - fv \right) / \left( \text{rms}(C\partial \tilde{\rho}'/\partial x) + \text{rms}(fv) \right) \right], \tag{48a}$$

$$\text{hydro. imbal} \quad = \quad \text{rms}\left[ \left( C\frac{\partial \tilde{\rho}'}{\partial z} - b' \right) / \left( \text{rms}(C\partial \tilde{\rho}'/\partial z) + \text{rms}(b') \right) \right], \tag{48b}$$

where rms indicates the root-mean-squared value of the quantity in brackets over the domain. The fields $\tilde{\rho}'$, $v$, and $b'$ are filtered before computing these diagnostics by removing scales (i) below 100km (to give the solid lines in panel d), (ii) below 10km (to give the dashed lines), and (iii) below 1km (i.e. unfiltered, to give the dotted lines). This gives us an indication of how the degree of imbalance is affected by scale.

There are variations of upward and downward vertical motion over the domain (Fig. 6a), but there are no regions that are specifically more convectively active than others. The $bu_{\text{eff}}$ diagnostic is fairly uniformly small over most of the domain (panel b) but does have more variability in the uppermost $5\,\text{km}$ of the domain where it is weakly negative in a thin layer at $14\,\text{km}$ (sandwiched between two strongly stable layers) during this snapshot. There is a small amount of disturbance of the tracer field after 3 hours (panel c).

The Rossby number is estimated to be small ($Ro \sim 0.06$), and the geostrophic imbalance is found to be moderate for the reference run (panel d), which stabilises to around 0.45 when only large scales are present, but higher, to around 0.7 to 0.8 when smaller scales are included (see the blue lines and the left-hand scale on panel d). The hydrostatic imbalance also increases as the scales shorten (see the red lines and the right-hand scale on panel d), but is much lower than the geostrophic imbalance (0.025 for the smallest scales). By estimating the magnitudes and length-scales of the fields in this run, the scale analysis in Sect. 2.2 does show that the last two terms in Eq. (17a) and in Eq. (17c) to be much larger than the other terms by about three and six orders of magnitude respectively.

In order to check that the linear analysis of Sect. 4 is relevant to the non-linear model integration, Table 3 compares the horizontal gravity wave speeds found from the linear analysis applied to the reference values, to the propagation speed of anomalies in the horizontal divergence field found from time sequences of model output (not shown; we refer to this as 'feature tracking'). Horizontal divergence, $\nabla_{\text{h}}(u,v) = \partial u/\partial x + \partial v/\partial y = \partial u/\partial x = \nabla_{\text{h}}^2 \chi$ (where $\chi$ is the velocity potential used in Sect. 4.1), is often associated with gravity wave activity. As is shown in Sect. 4, the wave speeds are dispersive in this system as so are a function of the horizontal and vertical wavenumbers. The relevant wavenumbers are found by looking at the characteristic length-scales of the fields (second and third columns in Table 3), which for all experiments conducted in this paper correspond respectively to horizontal wavenumber index $n_x = 3$, and vertical wavenumber index $n_z = 2$. The corresponding horizontal

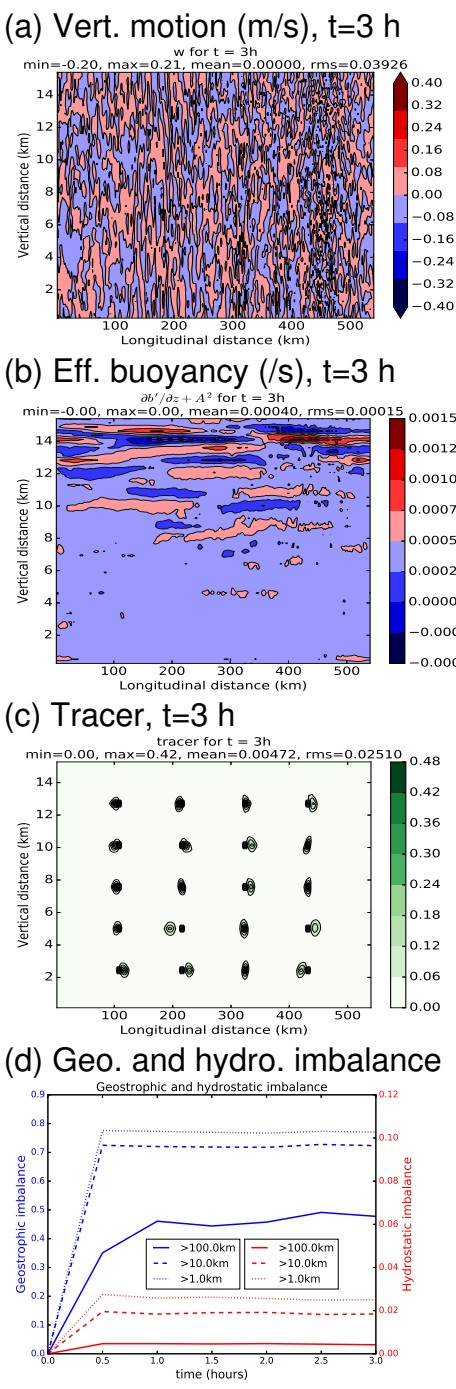

**Figure 6.** Selection of diagnostic fields for the reference parameters $A = 0.02\,\mathrm{s}^{-1}$, $B = 0.01$, $C = 10000\,\mathrm{m}^2\mathrm{s}^{-2}$. Panels (a-c) are after a three-hour forecast, except for the small rectangular shapes in (c), which represent the tracer at $t = 0$. In (d) the imbalances are shown as a function of forecast lead time and horizontal scale. The blue lines (and the left scale) are for geostrophic imbalance, and the red lines (and the right scale) are for hydrostatic imbalance. The Rossby number is $Ro \sim 0.06$.

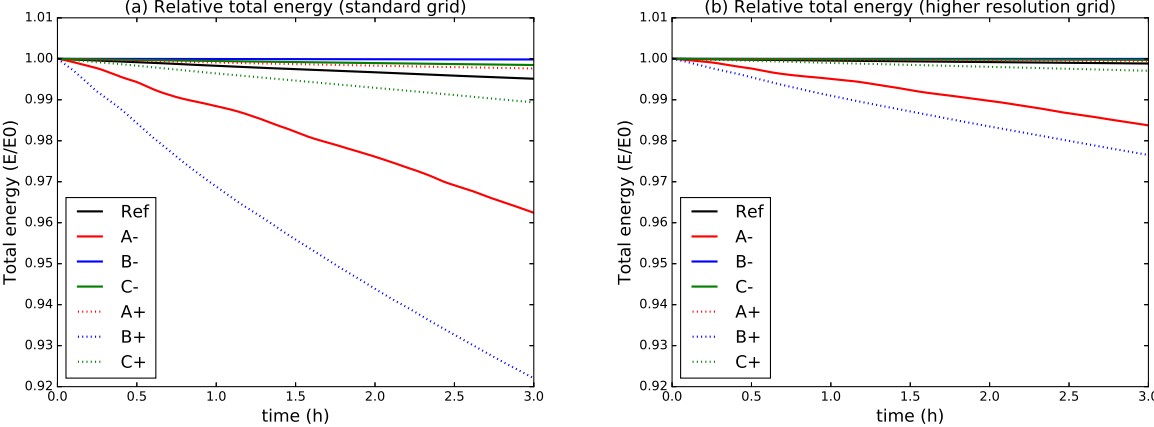

**Figure 7.** Relative total energy, $E(t)/E(0)$, from three-hour runs of the model with the reference parameters ("Ref"), and the six subsequent ways that the parameters were changed. The labels describe which parameter is modified in a model run and the + (-) indicates that it has been increased (decreased) by an order of magnitude from its reference value (with the remaining parameters unchanged). Panel (a) is for the model with the standard model grid-length as documented in this paper (e.g. $\Delta x = 1.5$ km) and panel (b) is for the same parameter sets and (down-scaled) initial conditions but with a model version with half grid-lengths in the horizontal and vertical directions simultaneously.

gravity wave speeds from the linear analysis are given in the fourth column, and the measured speed from the feature tracking is given in the fifth column. The values are comparable (in this and the other experiments), suggesting that the linear analysis is indeed a useful guide to the behaviour of the non-linear model.

Energy in the continuous system of equations was proven to be exactly conserved in Sect. 2.3, but the discretisation and numerical integration scheme introduces errors which will lead to non-conservation. Fig. 7a (black solid line) shows that these errors do lead to a small loss of energy over the three hours (less than half of a percent of the initial energy), which we assume is acceptable.

### 5.3.2 Changes to the parameter $A$

Recall that the parameter $A$ controls the gravity wave frequency and speed. In this section two three-hour integrations are done: one with $A$ decreased by an order of magnitude (A-, Fig. 8, left panels), and one with $A$ increased by an order of magnitude (A+, Fig. 8, right panels).

A- appears to result in more active $w$ values than in the reference run, and A+ appears to result in slightly less active $w$ values (panel a). The effective buoyancy (panel b) has more structure than in the reference run, with bands of lowered $bu_{\mathrm{eff}}$ appearing in A- (with patches of slightly negative $bu_{\mathrm{eff}}$ in the lower part of the domain which are too small to show as contours in the left plot of panel b), while A+ has no negative values at all. The increased vertical motion in A- is seen in the tracer fields (panel c), which have been transported more vertically in A- and slightly less vertically in A+ than the reference run. These results make physical sense given that $A$ controls the static stability of the fluid.

There are differences and similarities in the geostrophic and hydrostatic imbalances between A-, A+, and the reference run (panel d). In terms of geostrophic balance, A- is seen to be more balanced, and A+ is slightly less balanced than the reference run, and they all have similar Rossby numbers $\sim 0.06$. This observation seems counter-intuitive since we would expect the higher gravity wave speed of A+ to lead to a state of more, not less balance (although the adjustment time-scale – indicated by the continuous blue curves in Figs. 6d and 8d – is shorter for A+). The effect of $A$ on the gravity wave speeds is discussed in Sect. 4, and is confirmed further in Table 3, which compares the linear analysis and feature tracking analysis approaches. A- does indeed show a significant decrease in the gravity wave speed, and A+ shows an increase, and the degree of change is roughly consistent between the two approaches. In terms of hydrostatic balance, A- is slightly less balanced and A+ is much less balanced than the reference run. A- is consistent with our expectation, but A+ is not.

Even though we have ensured that the Courant number is small for all experiments (Sect. 4.7), we ask the question: is it possible that these unexpected balance results are artifacts of numerical imprecision? Is it possible that, as $A$ is changed by an order of magnitude for A- and A+, the changes in speeds of the waves (especially gravity), or changes in the vertical transport may result in a loss of precision, and hence lead to unreliable balance diagnostics?

The first issue (imprecision due to the change in the wave speeds, and thus the Courant number) is first studied by investigating the dependence of the balance diagnostics to changes of the integration time step $\Delta t$. It is found that $\Delta t$ can be increased up to $\sim 50$ times (i.e. to $\sim 5$ s, thus increasing the Courant number) before there are any noticeable changes to any of the balance diagnostics (not shown). This suggests that any loss of precision due to a too-long $\Delta t$ used in the main experiments can be ruled out. The Courant number can also be increased by reducing the grid size. It was found that the grid size can be halved (in the horizontal and vertical directions simultaneously) without qualitatively changing the balance results. In these higher resolution experiments it is found (not shown) that the measured degree of imbalance reduces in all runs, but the degree of imbalance relative to each experiment is unchanged (i.e. A- still has more geostrophic balance than the reference run, and A+ has slightly less, and this qualitative agreement is true also for the experiments to be discussed in later sections when run at higher resolution). This suggests that, although the higher-resolution runs have a different precision to the standard resolution runs, the fact that the results are qualitatively the same indicates that our results are probably robust.

The second issue (imprecision due to change in the vertical motion) is studied by noticing that there is an association between greater vertical motion, and increased energy loss (we assume that energy loss is in turn associated with imprecision). The less statically stable A- run results in more vertical motion and more energy loss than the reference run (4% loss over the three hours, solid red line in Fig. 7a) and the more stable A+ run results in less vertical motion and less energy loss (0.2% loss, dotted red line). Can this mean that variations in imprecision lead to unreliable diagnostics and hence are responsible for the counter-intuitive balance results summarised above? The higher-resolution runs mentioned above are more successful at conserving energy (the higher-resolution counterparts of Fig. 7a are shown on the same scale in panel b). A-, e.g. loses $\sim 2.5$ times less energy at the higher-resolution than A- at standard resolution (solid red lines). This suggests that the numerical integrations (and hence the balance diagnostics) for the higher-resolution runs are more reliable. Since the high-resolution balance diagnostics are qualitatively the same as those of the standard resolution results shown, this suggests that our results are not anomalous. This is not a definitive conclusion, but it does demonstrate some robustness of our results. The counter-intuitive balance results

though remain unexplained at this stage, although it should be noted that the balance diagnostics themselves may only be meaningful to compare within the same system of parameter values rather than between different systems.

### 5.3.3 Changes to the parameter $B$

Recall that the parameter $B$ (with $C$) controls mainly the acoustic wave speed. Two further three-hour integrations are done: one with $B$ decreased by an order of magnitude (B-, Fig. 9, left panels), and one with $B$ increased by an order of magnitude (B+, Fig. 9, right panels).

B- results in similar magnitude $w$ values as the reference run, and B+ results in a slight increase, but there is little change to the structures of the $w$ field (panel a). The effective buoyancy is largely unaffected by the changes in $B$ (panel b). A similar story applies to the tracers for B-, but the tracers for B+ do show increased vertical transport (panel c), which is consistent with the larger root-mean-squared $w$ for B+. B- (B+) has slightly more (less) geostrophic and hydrostatic balance, and the Rossby number remains small, $\sim 0.07$ ($\sim 0.11$). As discussed in Sect. 5.3.2, one would normally assume that a faster wave speed, as in the B+ run, would result in more balanced fields, but this is not the case here. The scale analysis in Sect. 2.2 reveals though that it is the product $BRo$, rather than just $Ro$ that is the quantity that scales terms that knock the system out of balance, and $BRo$ is smaller (larger) in B- (B+) than in the reference run.

Changing $B$ affects mainly the acoustic wave speed, but it can effect the gravity wave speed too. This is shown by the linear analysis and feature tracking analysis in Table 3, which shows a consistent decrease in the horizontal gravity wave speed for B- and an increase for B+. Although this effect of $B$ on the gravity wave speeds is large at these scales, its effect on the acoustic waves is much greater (for the scales in Table 3, the acoustic waves vary between 0.1 m/s for B- to 30 m/s for B+). Note that for smaller vertical scales the influence of $B$ on the gravity wave speed is much smaller.

Changing the $B$ parameter has a dramatic effect on the energy conservation (Fig. 7a), where B- yields the least energy loss of all experiments (on the scale used, the numerical loss of energy is indistinguishable from a perfectly conservative scheme – solid blue line in Fig. 7a), but B+ results in one of the most erroneous runs (an eight percent loss in energy over three hours, dotted blue line). Reducing the time step of the integration to one tenth of the value used in the main runs improves the energy conservation only marginally (not shown), but halving grid lengths (in the horizontal and vertical) improves the conservation (e.g., just over two percent loss over three hours for B+, dotted blue line in Fig. 7b).

### 5.3.4 Changes to the parameter $C$

Recall that the parameter $C$ (with $B$) controls mainly the acoustic wave speed. Two further three-hour integrations are done: one with $C$ decreased by an order of magnitude (C-), and one with $C$ increased by an order of magnitude (C-). The initial conditions for C- and C+ each differ from those used before as the procedure used to generate balanced initial conditions described in Sect. 5.2 from UM data depends on parameter $C$.

The C- and C+ results are not shown because they are virtually indistinguishable from the B- and B+ runs respectively (including the relative balance results). There are two differences though. The first is that $\tilde{\rho}'$ is scaled by $C^{-1}$ (when $C$ is decreased (increased) by an order of magnitude, $\tilde{\rho}'$ (not shown) is increased (decreased) by an order of magnitude compared

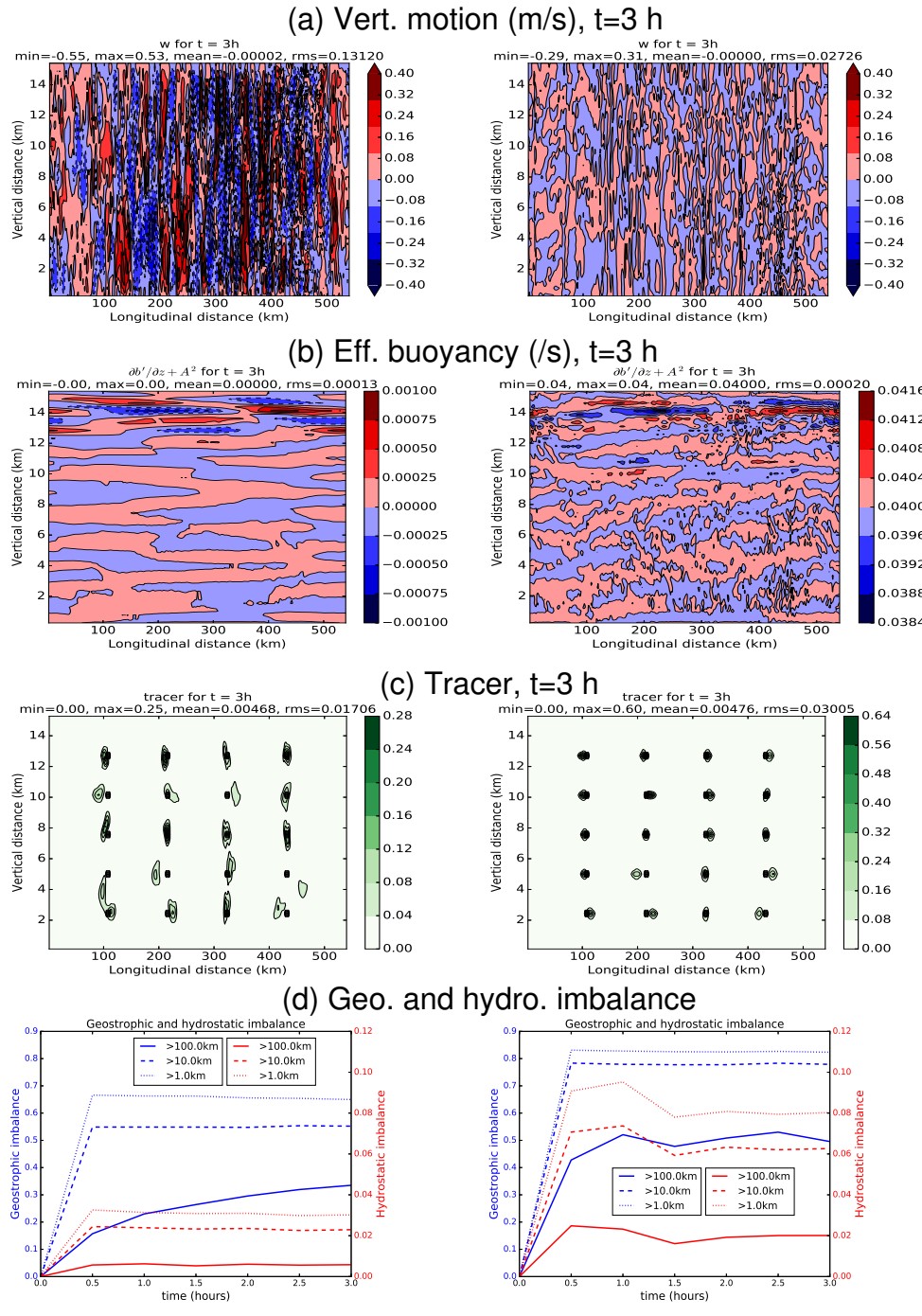

**Figure 8.** As Fig. 6 but for the modified $A$ parameter: $A = 0.002\,\mathrm{s}^{-1}$ (A-, left panels, $Ro \sim 0.06$), and $A = 0.2\,\mathrm{s}^{-1}$ (A+, right panels, $Ro \sim 0.07$). The remaining parameters are as for the reference run ($B = 0.01$, $C = 10000\,\mathrm{m}^2\mathrm{s}^{-2}$).

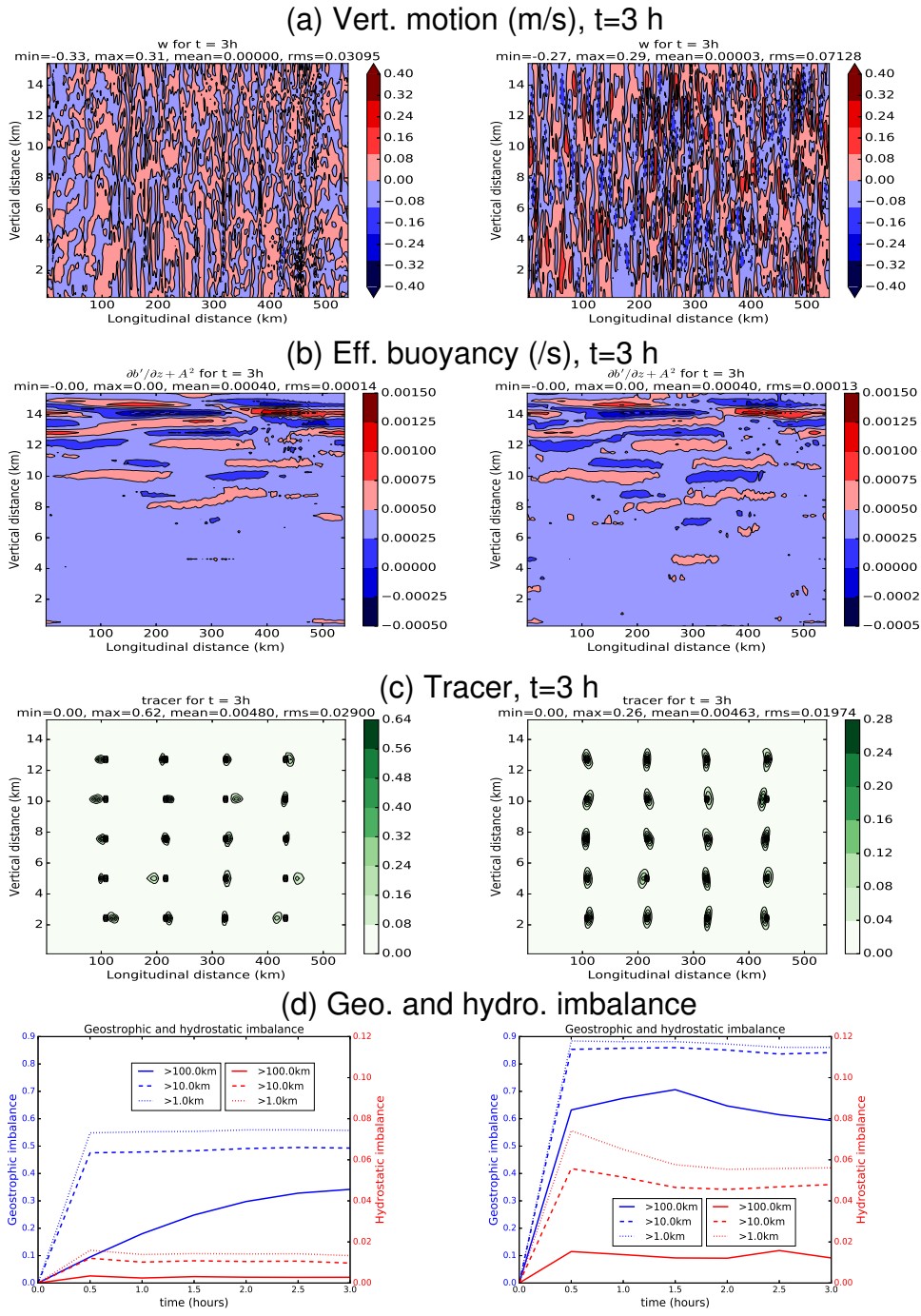

**Figure 9.** As Fig. 6 but for the modified $B$ parameter: $B = 0.001$ (B-, left panels, $Ro \sim 0.07$), and $B = 0.1$ (B+, right panels, $Ro \sim 0.11$). The remaining parameters are as for the reference run ($A = 0.02\mathrm{s}^{-1}$, $C = 10000\mathrm{m}^2\mathrm{s}^{-2}$).

to the B- (B+) runs, with the field structures remaining the same). This is seen in the scale analysis equations (17), where $C$ and $\mathcal{P}'$ (the characteristic value of $\tilde{\rho}'$) always appear together as a product. This is how the C- and C+ runs maintain the same level of geostrophic and hydrostatic balances as the B- and B+ runs respectively. The second difference is seen in the numerical scheme's energy loss. The B+ run (at the original model resolution) loses energy significantly (eight percent), but the C+ run loses less (one percent, dotted green line of Fig. 7a), and the C- run loses even less (around 0.1 percent, solid green line). This is another beneficial effect of introducing the $B$ parameter (i.e. the same value of a particular desired acoustic wave speed $\sqrt{BC}$ can be achieved by decreasing $B$ and increasing $C$).

## 6  Conclusions

A set of simplified mass and energy conserving equations have been derived which allow control of the gravity and acoustic wave characteristics to be controlled with three parameters, $A$ (the pure gravity wave frequency), $B$ (the modulation of the divergent term in the continuity equation, and of the advection terms in other equations), and $C$ (the proportionality constant for the toy model's equation of state). The term $\sqrt{BC}$ is the pure small-scale acoustic wave speed. The introduction of $B$ allows the acoustic wave speed to be reduced so that it is comparable to the gravity wave speed, hence allowing explicit integration schemes to be used to approximate the solution of the equation set (such as the split explicit, forward-backward scheme used here).

The linearised equations support a zero frequency Rossby-like mode and dispersive gravity and acoustic modes. The system is shown to behave in a way that reflects aspects of the atmosphere, namely geostrophic adjustment, convective behaviour influenced by buoyancy, and scale-dependent geostrophic and hydrostatic imbalances. Some of the results concerning the effect of changing $A$ on the degree of balance are counter-intuitive. Although we believe that the numerical results are robust, further work could be done to understand these results. It may be simply that the diagnostics (Eqs. (48)) cannot be compared between sets of parameters. The model has no water vapour, which simplifies the scheme considerable (although water vapour and moist processes could be added if required). The energy is not perfectly conserved in practice, which is due to the finite discretisation of the model (and to the choice of integration scheme), although numerical energy loss is assumed to be acceptable in most runs.

The purpose of developing this model is to facilitate research into ways of modelling the background error covariance matrix (**B**) used in convective-scale data assimilation. The **B**-matrix is normally modelled with guidance from large-scale dynamics, namely that geostrophic balance is dominant, and hydrostatic balance is exact. These assumptions are probably not applicable at convective-scales (as shown by Berre (2000); Bannister et al. (2011); Vetra-Carvalho et al. (2012); Bannister (2015), and as we have seen here, where more imbalance is present at the smaller scales). A key idea which will be explored in a forthcoming paper is to use the normal mode structure of the linearised equations to define the **B**-matrix rather than relying on imposed balances. It is hoped that this will have physically appropriate structures and the correct degree of balance at different scales (preliminary work has been done by Petrie (2012)).

## 7 Code and data availability

The model is written in Fortran-90, and the plotting code is written in python. This software is open-source and freely available on a Git Hub repository (Petrie et al., 2017). The initial conditions used to start the model runs studied in this paper is available from the same repository.

## 5 Appendix A: Linearisation

Many of the model's equations are non-linear, and are therefore difficult to analyse. Linearisation of each of these equations simplifies it to a form that neglects terms that are of second order or above. Each resulting linear equation is an approximate form of the non-linear equation, and is assumed to be a reasonable approximation for sufficiently small perturbations from a linearisation state. By way of example, (8f) is linearised here.

10     In (8f) there are three model variables: $p$, $\rho$, and $\theta$. These may be written as a (constant) linearisation state plus a perturbation, i.e. $p(x,z,t) = p_0(z) + p'(x,z,t)$, $\rho(x,z,t) = \rho_0(z) + \rho'(x,z,t)$, and $\theta(x,z,t) = \theta_\mathrm{R} + \theta_0(z) + \theta'(x,z,t)$, as in Eqs. (2) and (3). The subscript 0 indicates the linearisation state, except here for $\theta$, where the linearisation state is $\theta_\mathrm{R} + \theta_0(z)$. The linearisation state is assumed to obey Eq. (8f), namely $p_0 = \rho_0 R (p_0/p_{00})^\kappa (\theta_\mathrm{R} + \theta_0(z))$.

    Substituting these forms into Eq. (8f), and dropping the space and time co-ordinates for simplicity gives

$$15 \quad p_0 + p' = (\rho_0 + \rho') R \left( \frac{p_0 + p'}{p_{00}} \right)^\kappa (\theta_\mathrm{R} + \theta_0 + \theta').$$

Expanding this, applying the binomial theorem to the bracket raised to the power of $\kappa$, and ignoring products of perturbations gives

$$p_0 + p' \approx R \left( \frac{p_0}{p_{00}} \right)^\kappa \left( \rho_0(\theta_\mathrm{R} + \theta_0) + \rho_0 \theta' + \frac{\rho_0(\theta_\mathrm{R} + \theta_0)\kappa}{p_0} p' + (\theta_\mathrm{R} + \theta_0)\rho' \right).$$

Eliminating the reference state from each side (since it satisfies Eq. (8f)), assuming that $\theta_\mathrm{R} + \theta_0 \approx \theta_\mathrm{R}$, and rearranging, results 20 in the linearised equation (9).

    In this paper all equations of the ABC model (15) are linearised in a similar way to give the linearised model equations in (40), which are used to understand the structure of low amplitude perturbations (waves) that the system can permit. In this work the equations are linearised about a state of rest, i.e. $u_0 = 0$, $v_0 = 0$, and $w_0 = 0$.

## Appendix B: Conservation of mass and energy

25 Gauss' divergence theorem expressed generically is

$$\int_V \nabla \cdot \mathbf{A} \, dV = \oint_S \mathbf{A} \cdot d\mathbf{s}, \tag{B1}$$

(e.g. (Boas, 2006)) where $\mathbf{A}$ is an arbitrary vector field, $V$ is a volume existing in the vector field ($dV$ is a volume element of $V$), and $S$ is the surface of the volume ($d\mathbf{s}$ is an area element of $S$ multiplied by a unit vector pointing normal and outward to

the surface at the position of $d\mathbf{s}$). The divergence theorem may be used in the ABC model to prove conservation. The 'volume' $V$ represents the entire model domain, and the 'surface' $S$ therefore represents the boundary of the model (in a plane the divergence theorem reduces to an area integral on the left hand side and a closed line integral on the right hand side, and is equivalent to Green's theorem). The meaning of $\mathbf{A}$ depends upon the application.

To prove conservation of mass let $\mathbf{A} = \rho\mathbf{u}$, in which case (B1) gives:

$$
\int\limits_{x=0}^{L_x}\int\limits_{z=0}^{L_z} \nabla\cdot(\rho\mathbf{u})dxdz = \int\limits_{z=0}^{L_z} \rho(0,z)(-u(0,z))dz + \int\limits_{x=0}^{L_x} \rho(x,L_z)w(x,L_z)dx
$$
$$
+ \int\limits_{z=L_z}^{0} \rho(L_x,z)u(L_x,z)(-dz) + \int\limits_{x=L_x}^{0} \rho(x,0)(-w(x,0))(-dx), \tag{B2}
$$

where $L_x$ and $L_z$ are the length and height of the model respectively, and the four terms on the right hand side represent contributions from the four sides of the model's domain (starting from the lower-most/west-most point and integrating clockwise

around the domain). The model has zero vertical wind values at the top and bottom boundaries (Table 2), which removes the second and fourth terms on the right hand side. Furthermore swapping the integration limits on the third integral (which introduces a minus sign to that term), and noting that the fields are periodic in the horizontal ($\rho(L_x,z)u(L_x,z) = \rho(0,z)u(0,z)$) leads to the first and third terms cancelling. The right hand side is therefore zero. The equation describing the evolution of the total mass is

$$
\frac{\partial\text{total mass}}{\partial t} = \frac{\partial}{\partial t}\left(\int\limits_{x=0}^{L_x}\int\limits_{z=0}^{L_z}\rho dxdz\right)
$$
$$
= \int\limits_{x=0}^{L_x}\int\limits_{z=0}^{L_z}\frac{\partial\rho}{\partial t}dxdz
$$
$$
= -B\int\limits_{x=0}^{L_x}\int\limits_{z=0}^{L_z}\nabla\cdot(\rho\mathbf{u})\,dxdz = 0,
$$

where the last line follows from the second line using the mass continuity equation given in Sect. 2.3.1, and the final zero results from Eq. (B2) being zero as shown above. This proves conservation of total mass in the ABC model.

To prove conservation of energy, the second and third terms in (29) must be shown to be zero. For the second term use Gauss' divergence theorem with $\mathbf{A} = B(E_k + E_b)\mathbf{u}$, and for the third term use $\mathbf{A} = C(\tilde{\rho}'\tilde{\rho}\mathbf{u})$. Since each of these is proportional to $\mathbf{u}$, the same arguments used for mass conservation can be used to show that the second and third terms of (29) do not contribute. This proves that the rate of change of total energy integrated throughout the domain, as in (29) is zero, and therefore energy is conserved.

*Author contributions.* This work has emerged from the doctoral thesis of REP Petrie (2012). MJPC provided the main scientific guidance for the particular simplifications used to define the toy model, and REP did most of the coding and testing under the supervision of RNB.

REP performed the initial model runs for her thesis, and RNB developed the code further and performed the model runs for this paper. This manuscript was drafted by RNB, and REP, with substantial advice from MJPC.

*Competing interests.* The authors declare that they have no conflict of interest.

*Disclaimer.*

5 *Acknowledgements.* REP was supported by a NERC/NCEO studentship for the course of this work, RNB is supported by the NCEO, and MJPC is supported by the Met Office. REP and RNB would like to thank the local assistance from Terry Davies, who helped REP code the integration scheme, and Sue Ballard for general Met Office supervision. The authors would like to also thank the two anonymous reviewers for their careful reviews and useful comments.

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

**Table 1.** Description of the main symbols used in this work.

| Symbol | Description | Symbol | Description |
|---|---|---|---|
| $x$ | Longitudinal position | $z$ | Height |
| $t$ | Time | | |
| $u$ | Zonal wind | $v$ | Meridional wind |
| $w$ | Vertical wind | $\mathbf{u}$ | 3D wind vector |
| $\rho$ | Density | $p$ | Pressure |
| $\theta$ | Potential temperature | $\theta_{\mathrm{R}}$ | $z$-independent part of ref. $\theta$ (3) |
| $\tilde{\rho}$ | Scaled density (14) | $b$ | Buoyancy (7) |
| $q$ | Passive tracer concentration | | |
| $\rho_0, p_0, \theta_0, \tilde{\rho}_0, b_0$ | Reference state variables (2) | $\rho', p', \theta', \tilde{\rho}', b'$ | Perturbation quantities (2) |
| $\mathcal{U}, \mathcal{V}, \mathcal{W}, \mathcal{P}', \mathcal{B}'$ | Characteristic values $u, v, w, \tilde{\rho}', b'$ | $u^*, v^*, w^*, \tilde{\rho}'^*, b'^*$ | Variables scaled by charac. values |
| $\mathcal{L}_u^{\mathrm{H}}, \mathcal{L}_v^{\mathrm{H}}, \mathcal{L}_w^{\mathrm{H}}, \mathcal{L}_{\tilde{\rho}'}^{\mathrm{H}}, \mathcal{L}_{b'}^{\mathrm{H}}$ | Characteristic horiz. length-scales | $x_u^*, x_v^*, x_w^*, x_{\tilde{\rho}'}^*, x_{b'}^*$ | Long. pos. variables scaled by horiz. length-scales |
| $\mathcal{L}_u^{\mathrm{V}}, \mathcal{L}_v^{\mathrm{V}}, \mathcal{L}_w^{\mathrm{V}}, \mathcal{L}_{\tilde{\rho}'}^{\mathrm{V}}, \mathcal{L}_{b'}^{\mathrm{V}}$ | Characteristic vert. length-scales | $z_u^*, z_v^*, z_w^*, z_{\tilde{\rho}'}^*, z_{b'}^*$ | Height variables scaled by vert. length-scales |
| | | $t^*$ | Time variable scaled by time-scale |
| $g$ | Acceleration due to gravity | $f$ | Coriolis parameter |
| $R$ | Gas constant for dry air | $\kappa$ | Ratio of specific heats |
| $c_p$ | Specific heat capacity (constant $p$) | $p_{00}$ | 1000 hPa |
| $\mathbf{k}$ | Vertical unit vector | $N$ | Brunt-Väisälä frequency |
| $Ro$ | Rossby number | | |
| $A$ | Pure gravity wave frequency | $B$ | Modulation of the divergent term in the continuity equation |
| $C$ | Proportionality constant for toy model equation of state (13) | | |
| $E_{\mathrm{k}}$ | Kinetic energy (22) | $E_{\mathrm{b}}$ | Buoyant energy (25) |
| $E_{\mathrm{e}}$ | Elastic energy (27) | $E$ | Total energy |
| $\chi, \psi$ | Velocity potential and streamfunction (39) | $\Delta t, \delta t$ | Time step and sub time step of integration scheme |
| $k, m$ | Horiz., vert. wavenumbers | $n_x, n_z$ | Horiz., vert. wavenumber indices |
| $\sigma, \sigma_{\mathrm{R}}, \sigma_{\mathrm{g}}, \sigma_{\mathrm{g}'}, \sigma_{\mathrm{a}}, \sigma_{\mathrm{a}'}$ | Wave frequency, specifically Rossby, gravity, acoustic | $L_x, L_z$ | Horiz., vert. domain sizes |
| $c_{\mathrm{g}}, c_{\mathrm{a}}$ | Gravity and acoustic group speeds | $bu_{\mathrm{eff}}$ | Effective buoyancy |

**Table 2.** Upper and lower boundary conditions of each prognostic model variable, $z = 0$ is the the lower boundary position and $z = L_z$ is the upper boundary position.

|  | Lower | Upper |
|---|---|---|
| $u$ | $u(0) = 0$ | $\frac{\partial u(L_z)}{\partial z} = 0$ |
| $v$ | $v(0) = 0$ | $\frac{\partial v(L_z)}{\partial z} = 0$ |
| $w$ | $w(0) = 0$ | $w(L_z) = 0$ |
| $\tilde{\rho}'$ | $\frac{\partial \tilde{\rho}'(0)}{\partial z} = 0$ | $\frac{\partial \tilde{\rho}'(L_z)}{\partial z} = 0$ |
| $b'$ | $b'(0) = 0$ | $b'(L_z) = 0$ |

**Table 3.** Summary of the characteristic horizontal and vertical length-scales found for each of the experiments (estimated after 3 hour model integrations), the horizontal gravity wave speeds corresponding to these scales as found from the linear analysis, and an estimate of the gravity wave speed found by tracking features in the horizontal divergence field ($\partial u/\partial x$).

| Experiment | $\mathcal{L}_v^{\mathrm{H}}$ (km) | $\mathcal{L}_v^{\mathrm{V}}$ (km) | Horiz. grav. wave speed, $c_{\mathrm{g}}$ (m/s) | |
|---|---|---|---|---|
| | | | Linear analysis | Feature tracking |
| Ref | 97 | 4.6 | 8.6 | 9.6 |
| A- | 99 | 4.5 | 1.3 | 2.7 |
| A+ | 99 | 4.7 | 9.4 | 10.1 |
| B- | 100 | 4.6 | 2.1 | 2.9 |
| B+ | 98 | 4.7 | 18.6 | 22.1 |