# Peer review of "The "ABC model": a non-hydrostatic toy model for use in convective-scale data assimilation investigations"

_Geoscientific Model Development, 2017_

## Referee Comment (RC1) · Anonymous Referee #1 · 22 Jun 2017

Review of the manuscript under the title "The "ABC model" (Vn 1.0) : a non-hydrostatic toy model for use in convective-scale data assimilation" by Ruth Elisabeth Petrie, Ross Noel Bannister and Michael John Priestley Cullen submitted to Geoscientific Model Development (MS No. gmd-2017-68).

Overview

The manuscript introduces a new "toy model" (abbreviated as "ABC model") suitable for modelling on convective scales that is based on a simplified system of intermediate complexity derived from the 3-D Euler equations. The "ABC model" is designed to meet the following requirements of a "toy model" environment: 1) "toy model" is cheap

to integrate, namely it can be integrated explicitly with a not too small time-step (a split explicit forward-backward scheme is used in this study) 2) "toy model" is inexpensive to handle (the model is intended to be run in vertical slice geometry) 3) "toy model" mimics a multi-scale behaviour of the real atmosphere (the model retains geostrophic and hydrostatic balances on large scales and allows intermittent convection-like behaviour on the small scales) 4) "toy model" is flexible (the stiffness of the system is controlled by tunable parameters at the same time as the important characteristics of the system, such as the conservation of total energy, are preserved )

The authors describes in a clear and well-motivated pedagogical way the philosophy behind the simplifications and assumptions which were done to the complex 3-D system in order to fulfil the concept of a "toy model" environment. The name of the model originates from three tunable parameters A,B and C. They are introduced into the model in order to allow a control on the speed of dispersive waves (gravity waves and acoustic waves). This becomes possible due to a linearized and simplified equation of state used in the "toy model" that allows a much weaker coupling between the acoustic and gravity waves than the original non-linear equation of state imposes. Parameter A controls the gravity wave frequency, parameter B provides the modulation of the divergent term in the continuity equation and parameter C defines the compressibility of the air. By optimally choosing A, B and C parameters it is possible to obtain a state where acoustic and gravity waves have a comparable speed. This allows to choose an affordable time step when an inexpensive explicit time integration scheme is employed. The authors demonstrate that these tunable parameters are designed so that the important properties of the system are preserved in the "toy model" version. Namely the flow is in a close to hydrostatic and geostrophic balance on large scales and the total energy of the flow is conserved. In order to illustrate a typical behaviour of the "ABC model", the authors propose 4 kinds of diagnostic quantities (a vertical wind speed used as an indication of convection-like motion; an effective buoyancy and a tracer distribution used to illustrate flow stability; and the degree of relative geostrophic and hydrostatic imbalances). The authors conduct a number of experiments to demonstrate the sensitivity

of these diagnostic quantities to the different values of the A, B and C parameters and explain the reference choice. To get a better insight into a meaning of these three parameters a linear analysis of the "ABC" model is performed decomposing the flow into normal modes. The authors motivate the development of this model as a first necessary step needed in order to progress with design of convective scale data assimilation systems. The next step will be to use the normal mode structure of the linearized equations of the "ABC model" to derive the background error covariance matrix suitable at meso-scales.

The toy model uses a similar grid as that of the Southern UK version of UK Met Office's Unified Model (UM) with a resolution of 1.5 km on an Arakawa-C grid and a time step of 1s. The toy model is periodic in zonal direction, is homogeneous in meridional direction and uses 60 vertical model levels with a regular spacing of 250m. The model has flat orography. The upper and lower boundary conditions are defined to conserve the total mass and energy. The model has dry dynamics and relies on the wave breaking to drive a convection-like motion. The initial conditions for the simulations are taken from an output of UM model and recomputed imposing hydrostatic and geostrophic balances of the "ABC model".

General comments.

The paper is very well written and addresses one of the urgent areas in the development of the convective-scale data assimilation, namely the existence of a practical and a physically realistic environment. Such toy model framework as the authors propose would indeed be useful and almost necessary developing new data assimilation methodology.

The paper has a very clear structure of presentation that is easy to follow. First the toy model is derived, clearly communicating the set of physically based approximations that are made and then the set of a "toy model" simplifications that are introduced including the choice ot tunable parameters. Then the properties of the obtained toy

model are investigated in term of scale analysis and the conservation of total energy and mass (Section 2).

Then the numerical implementation of the model is described. Even though this part is based on the already published work (Cullen and Davies, 1991) and contains many technical details, the information is important in the context of this work in particular for the reproducibility of the results (Section 3).

Afterwards the normal mode decomposition of the underlying linear system is performed in order to provide a better insight into the properties of the system and to understand the meaning of the introduced parameters. The normal modes (gravity and acoustic waves) are studied both analytically and numerically including the dependency of the wave speed on the choice of parameters (Section 4)

Then the numerical simulations with the "ABC model" are performed including the systematic exploration of the model behaviour over the parameter space. Such questions like "how the degree of the imbalances, the precision of the numerical solution and the activity of the convection-like motion depend on different values of tunable parameters" are addressed (Section 5).

What I am missing here is a discussion that would relate findings from the Section 5 to findings in other Sections (2, 3, and in particular 4). Some of the results presented in the Section 5 are somewhat contra-intuitive and should be more elaborated. The authors describes mainly WHAT happens in terms of the selected diagnostic quantities when the parameters are changing. More insight is needed in WHY this happens. . .

For example, in Section 4 the authors motivate the choice of the time step (1s, with a sub-time step 0.5s) in the split explicit forward-backward scheme, which is a first order scheme, by the estimate of the highest frequency of the acoustic wave (subsection 4.6) corresponding to the reference parameters (A=0.02 s-1, B = 0.01, C=10000 m2 s-2). In section 4 the authors also analyse how the values of parameters A,B and C influence speed of gravity and acoustic waves (subsection 4.5). In Section 5 the authors conduct 3 pairs of the additional experiments with increased/decreased values of parameters A, B and C . It is not totally clear from the description of these experiments if some adjustments to the time step of integration were done due to change in the speed of waves propagation. The following questions emerge : 1) The choice of the reference parameters is such that the speed of the gravity and the acoustic waves becomes comparable. The change in parameters values will change the relative ratio between the speed acoustic and gravity waves. For example in A+ experiment acoustic waves will be slower than the gravity waves...What implication will this have on the precision of numerical solution? 2) Obviously the dissipation of energy for the energy conserving system is due to the lack of precision of the numerical solution. What is the main cause of the numerical error? 3) B+ experiment results in the largest error in the conservation of energy. This is the experiment with the largest non-linear term (advection) and with the highest horizontal speed for acoustic wave. B+ experiment results in the increased vertical transport. To what extend increased vertical transport is related to the numerical noise? 4) In all experiments higher values of A+, B+ or C+ (and higher speed for wave propagation) results in a more geostrophically imbalanced field (figures 6,8 and 9, panel d) to the right), while decreased values of parameters A-, B- and C- (and lower speed for wave propagation) all results in somewhat lower geostrophical imbalance that grows slower in time (figures 6, 8 and 9, panel d) to the left). At the same time all three parameters control different features of the flow (static stability, degree of non-linearity and compressibility ). What mechanism lies behind this effect? Can a too large time step produce a less balanced field?

More Specific comments

Section1. Introduction. – p2. l10. Sentence "The DA scheme that combines the observed and the background data should provide an analysis which is approximately consistent with the observations and the model." An expression "approximately consistent" is misleading in this context. Please reformulate the statement.

– p2. l30 . Sentence "These methods though suffer from noise in the sampled error

covariance matrix and so rely on fixes such as localisation, which is known to destroy balances when they are relevant". I think it is inappropriate to call localisation approach as "fixes" because this is a mathematically sound method that increase rank of the resulting matrix, even though it indeed might destroy balances. Please reformulate the statement or remove ". . . fixes such as ..."

– p3. l28 . Sentence "These simplifications permit a large-scale balanced flows and sporadic small-scale non-hydrostatic flows (i.e. convection) to coexist within the framework of a simplified and practical model." I do not think that it is appropriate to relate all sporadic small-scale non-hydrostatic flows to convection. Please moderate the statement. One of the biggest problems when modelling on high-resolution is to distinguish between a convective motion and a numerical noise that often happen on similar scales. The toy model environment such as the proposed one could be a very useful framework to study the propagation of the error on meso-scales that comes different sources (the error in the initial conditions, the model error due to deficiencies in the numerical scheme and the interaction of these two sources)

Section 2. Subsection 2.1.2 "The "ABC model" modifications" – p6 l9 . "Linearizing Eq. (8f) about the basic state . . .." Please explain explicitly the procedure of linearizing equation around the basic state. Some readers might not be familiar with this approach

Subsection 2.3.1 "Conservation of mass" – p8 l21 Please explain or provide the references to what "the divergence theorem" (known more as Gauss's theorem or Gauss-Ostrogradsky theorem in calculus) means

Subsection 2.3.6 "Total combined energy and its conservation" – p10 l5. Here the notation "the divergence theorem" is used again without any definition or explanation...

Section 4. Subsection 4.1"Linearization" – p13 l19-21 . "The non-linear model equations are linearized about the reference state and a state of rest. It is convenient to write the model equations in terms of velocity potential and streamfunction. The Helmholtz theorem gives: . . .." Please explain do what "the reference state" and "a state at rest"

mean, what does "the linearization of the equation" mean and why this procedure is performed; Please provide references or explain "the Helmholtz theorem"; To obtain equations (39) from equations (15) the flow was first split in divergent and rotational parts (15 a,b → 39 a,b) and then expressed in terms of velocity potential and stream-function. Equations 15 a,b,c,e were linearized aroud a state at rest and equation 15 d was linearised around the reference state

– p 14 Please explain more clearly in words what procedure is performed here and what is the meaning of the analysis in spectral space. For reader it might be difficult to follow derivations. Please explain the decomposition of the flow onto orthogonal modes.

Subsection 4.6 – p17 l 18. Please explain what is the meaning of Courant number and it is "sufficiently small"

Section 5 "The ABC model integration results"

Subsection 5.2."Intermittent convection-like behaviour"

– p20 l3 . "An obvious indication of presence of convection is vertical motion and . . . ". I think it is important to refer here to Eq 44 indicating that for pure linear systems the balance part of the flow does not have vertical wind component.

Subsection 5.3 "Systematic exploration of model behaviour over parameter space"

Figures 6, 8 and 9, panel d) The plots are very difficult to read, another line style/colours/symbols need to be used. Legends should be moved away because they destroy information on the plots.

Figure 7. It is difficult to read the plot. Another style/colours/symbols need to be used

Technical Comments

– p11 l16 . Should be "$\delta t=\Delta t/n$" instead of "$\delta t=\delta t/n$"

[Figure]

– p15 l22. Should be "$\sigma$a" instead of "$\sigma$g"

– p15 l24 . Should be "small-scale" instead of "scall-scale"

---

## Referee Comment (RC2) · Anonymous Referee #2 · 27 Jul 2017

- General comments:

This paper presents a 'toy' model for convective scale data assimilation. After physically-based simplifications and more modifications, the end result is a model with 3 tunable parameters which allows for large and small scale motions and yet is computationally non-expensive, and where the effects of acoustic and gravity waves are identifiable and traceable.

I think this work fills a very important gap for DA experiments (and beyond DA) in the sense that right now we have either extremely simple models or medium-complexity or full GCM's to use. As indicated in the introduction, one of the only toy models that

allow convective-type behaviour is that of Craig and Wursch. So this is a welcomed development indeed.

In my opinion, the structure of the paper is very clear, and the steps taken from the Navier-Stokes equations to the final product are very well explained. Caution has been taken in conservation of energy. The authors have also discussed the numerical implementation, as well as a linearised analysis, and finally some numerical experiments.

- Specific comments: Having read the other review, I strongly agree with the request for a further analysis of the sample integrations in section 5, and their relation with the findings of the previous sections. I have to requests not mentioned before. 1. Understandably, there are a lot of symbols. I believe it would be useful to have a list of symbols as an appendix. While reading the paper, I had to go back sometimes between sections to know the differences in variables, e.g. $p$ vs $p\_0$ vs $p\_{00}$, or when are the variables calligraphic, when do they have a star, etc. 2. It was a little difficult to read the axes in some of the figures. In fact, could some of the panels be done larger? For example, a figure has 4 panels stacked vertically with a lot of white space to the sides, while a 2x2 grid would show them better.

- Typos: There are some typos both in the text and in the equations. For instance $\delta t = \delta t /N$. These have been identified by the other reviewer so I am not repeating them.

---

## Author Comment (AC1) · 27 Aug 2017

[english]article [T1]fontenc [latin9]inputenc geometry verbose,tmargin=3cm,bmargin=3cm,lmargin=3.5cm,rmargin=3.5cm textcomp babel

**Response to referee 1**
We would like to thank referee 1 for his/her comments. The referee's comments that require attention to the paper are reproduced below preceded with "**Referee comment**", the authors' responses are preceded with "**Authors' response**", and our actions are preceded with "**Authors' changes**". We have produced a revised manuscript of the paper, but we understand that this is not meant to be uploaded with these comments. Only a brief explanation of the changes are given here and we hope that a revised manuscript will be requested where the detailed changes are given (in the text below we do refer to parts of the revised paper where any changes are made should a revised manuscript be requested).

- **Referee comment**: What I am missing here is a discussion that would relate findings from the Section 5 to findings in other Sections (2, 3, and in particular 4). Some of the results presented in the Section 5 are somewhat contra-intuitive and should be more elaborated. The authors describes mainly WHAT happens in terms of the selected diagnostic quantities when the parameters are changing. More insight is needed in WHY this happens. . .

    – **Authors' response**: Our purpose of the earlier sections of the paper was to understand how the parameters relate to different aspects of the system, and in order to choose suitable $A$, $B$ and $C$ parameters for the numerical work, as well as a suitable time step. Despite efforts to understand why the apparent degree of imbalance changes in the way it does as $A$ is modified, we have not been able to explain this (see penultimate change below though). We have increased the cross referencing to earlier sections.

– **Authors' changes**

* Some more description is now given in Sect. 4.7 regarding the choice of time step.
* There are also references back to previous sections from Sect. 5 as follows.
* In Sects. 5.3.1 (discussion of the reference parameters), 5.3.2 (changes of $A$), and 5.3.3 (changes of $B$) we now compare the gravity wave speeds found from the linear analsysis in Sect. 4 with the propagation speeds found in the non-linear model runs. This involved the inclusion of a new table showing that these speeds are consistent (Tab. 3).
* In Sect. 5.3.3 (discussion of changes to $B$), we do refer back to the scale analysis equations in Sect. 2.2.
* End of Sect. 5.3.2 (discussion of changes to $A$) and conclusions section, we comment on the possible (lack of) validity of using the balance diagnostics (48) to compare the degree of imbalance between different parameter values.
* Other related specific points are responded to below.

• **Referee comment**: For example, in Section 4 the authors motivate the choice of the time step (1s, with a sub-time step 0.5s) in the split explicit forward-backward scheme, which is a first order scheme, by the estimate of the highest frequency of the acoustic wave (subsection 4.6) corresponding to the reference parameters (A=0.02 s-1, B = 0.01, C=10000 m2 s-2). In section 4 the authors also analyse how the values of parameters A,B and C influence speed of gravity and acoustic waves (subsection 4.5). In Section 5 the authors conduct 3 pairs of the additional experiments with increased/decreased values of parameters A, B and C . It is

not totally clear from the description of these experiments if some adjustments to the time step of integration were done due to change in the speed of waves propagation. The following questions emerge :

– **Referee comment**: 1) The choice of the reference parameters is such that the speed of the gravity and the acoustic waves becomes comparable. The change in parameters values will change the relative ratio between the speed acoustic and gravity waves. For example in A+ experiment acoustic waves will be slower than the gravity waves...What implication will this have on the precision of numerical solution?

* **Authors' response**: This is correct and we had not completely accounted for this.
* **Authors' changes**: We have found the maximum wave frequency over all of the possible parameter values and have chosen a new suitable time step for the model which will be suitable for all runs made ($\Delta t = 0.1$s now, instead of $\Delta t = 1$s in the original). Please see revised Sect. 4.7.

– **Referee comment**: 2) Obviously the dissipation of energy for the energy conserving system is due to the lack of precision of the numerical solution. What is the main cause of the numerical error?

* **Authors' response**: Shortening the time step improves the conservation of energy only marginally, but having a smaller grid length improves conservation significantly.
* **Authors' changes**: This is now discussed in Sects. 5.3.2, 5.3.3, Fig. 7, and briefly in the conclusions.

– **Referee comment**: 3) B+ experiment results in the largest error in the conservation of energy. This is the experiment with the largest non-linear term (advection) and with the highest horizontal speed for acoustic wave. B+

experiment results in the increased vertical transport. To what extend increased vertical transport is related to the numerical noise?

  * **Authors' response**: There is indeed an association with vertical transport and imprecision. This raises the question whether changes in (im)precision will result in unreliable balance results (and hence possibly explain the counter-intuitive results). We do not believe that this is the case as the integrations at higher resolution result in dramatically less loss of energy, but the balance results are not qualitatively changed.
  * **Authors' changes**: This is discussed in the last part of Sects. 5.3.2, and 5.3.3.

– **Referee comment**: 4) In all experiments higher values of A+, B+ or C+ (and higher speed for wave propagation) results in a more geostrophically imbalanced field (figures 6,8 and 9, panel d) to the right), while decreased values of parameters A-, B- and C- (and lower speed for wave propagation) all results in somewhat lower geostrophical imbalance that grows slower in time (figures 6, 8 and 9, panel d) to the left). At the same time all three parameters control different features of the flow (static stability, degree of non-linearity and compressibility ). What mechanism lies behind this effect? Can a too large time step produce a less balanced field?

  * **Authors' response**: These results do indeed seem counter-intuitive, and we have still not been able to explain them, despite an in depth investigation.
  * **Authors' changes**: We have considered that the unexplained balance results may be due to imprecision (as above – either due to inability to support the wave motions in some of the runs, or the differing degrees of vertical motion – see Sect. 5.3.2). We believe tentatively that the results are not due to imprecision (by doing experiments with different grid-lengths), although we are as yet unable to explain the apparently

anomalous effect of $A$ on the degree of imbalance – again Sect. 5.3.2. As stated above a possible cause of these 'unexplained' results is that the balance diagnostics (48) may not be suitable for comparison between different parameter sets. See end of Sect. 5.3.2 (discussion of changes to $A$) and conclusions section. We do think though that the effect of $B$ though can be explained – Sect. 5.3.3.

• **Referee comment**: Section1. Introduction. – p2. l10. Sentence "The DA scheme that combines the observed and the background data should provide an analysis which is approximately consistent with the observations and the model." An expression "approximately consistent" is misleading in this context. Please reformulate the statement.

  – **Authors' response**: Thank you.
  – **Authors' changes**: This statement has been reformulated (Sect. 1).

• **Referee comment**: p2. l30 . Sentence "These methods though suffer from noise in the sampled error covariance matrix and so rely on fixes such as localisation, which is known to destroy balances when they are relevant". I think it is inappropriate to call localisation approach as "fixes" because this is a mathematically sound method that increase rank of the resulting matrix, even though it indeed might destroy balances. Please reformulate the statement or remove ". . . fixes such as ..."

  – **Authors' response**: Thank you.
  – **Authors' changes**: This statement has been reformulated (Sect. 1).

• **Referee comment**: p3. l28 . Sentence "These simplifications permit a large-scale balanced flows and sporadic small-scale non-hydrostatic flows (i.e. convection) to coexist within the frame- work of a simplified and practical model." I

do not think that it is appropriate to relate all sporadic small-scale non-hydrostatic flows to convection. Please moderate the statement. One of the biggest problems when modelling on high-resolution is to distinguish between a convective motion and a numerical noise that often happen on similar scales. The toy model environment such as the proposed one could be a very useful framework to study the propagation of the error on mesoscales that comes different sources (the error in the initial conditions, the model error due to deficiencies in the numerical scheme and the interaction of these two sources)

- **Authors' response**: Yes, we agree that not all small-scale motion is convection.
- **Authors' changes**: Have replaced "i.e. convection" with "including convection" (Sect. 1).

- **Referee comment**: Section 2. Subsection 2.1.2 "The "ABC model" modifications" – p6 l9 . "Linearizing Eq. (8f) about the basic state . . .." Please explain explicitly the procedure of linearizing equation around the basic state. Some readers might not be familiar with this approach

- **Authors' response**: Thank you.
- **Authors' changes**: Appendix A has been added to describe linearisation, with an example of linearising (8f) to give (9). The appendix is referenced from the main text in Sect. 2.1.2 (after (9)).

- **Referee comment**: Subsection 2.3.1 "Conservation of mass" – p8 l21 Please explain or provide the references to what "the divergence theorem" (known more as Gauss's theorem or Gauss- Ostrogradsky theorem in calculus) means

- **Authors' response**: Yes, thank you.

- **Authors' changes**: Appendix B has been added to show how the divergence theorem is used to show conservation. The appendix is referenced from the main text in Sect. 2.3.1. The divergence theorem is explained in Appendix B.

- **Referee comment**: Subsection 2.3.6 "Total combined energy and its conservation" – p10 l5. Here the notation "the divergence theorem" is used again without any definition or explanation...

- **Authors' response**: Thank you.
- **Authors' changes**: This is included in Appendix B.

- **Referee comment**: Section 4. Subsection 4.1"Linearization" – p13 l19-21 . "The non-linear model equations are linearized about the reference state and a state of rest. It is convenient to write the model equations in terms of velocity potential and streamfunction. The Helmholtz theorem gives: . . .." Please explain do what "the reference state" and "a state at rest" mean, what does "the linearization of the equation" mean and why this procedure is performed; Please provide references or explain "the Helmholtz theorem"; To obtain equations (39) from equations (15) the flow was first split in divergent and rotational parts (15 a,b → 39 a,b) and then expressed in terms of velocity potential and streamfunction. Equations 15 a,b,c,e were linearized around a state at rest and equation 15 d was linearised around the reference state

- **Authors' response**: Thank you.
- **Authors' changes**: The "reference state" text has been removed, and the meaning of "a state of rest", "linearisation" and the reason why linearisation is done is now explained in appendix A (and referenced from the main text at the start of Sect. 4.1). The Helmholtz theorem is now explained and referenced in Sect. 4.1.

- **Referee comment**: p 14 Please explain more clearly in words what procedure is performed here and what is the meaning of the analysis in spectral space. For reader it might be difficult to follow derivations. Please explain the decomposition of the flow onto orthogonal modes.

    – **Authors' response**: Thank you.
    – **Authors' changes**: Some extra text has been added at the start of Sect. 4.

- **Referee comment**: Subsection 4.6 – p17 l 18. Please explain what is the meaning of Courant number and it is "sufficiently small"

    – **Authors' response**: Thank you.
    – **Authors' changes**: Some extra text at the end of Sect. 4.7 has been added, including mention of the CFL condition.

- **Referee comment**: Subsection 5.2. "Intermittent convection-like behaviour" – p20 l3 . "An obvious indication of presence of convection is vertical motion and . . . ". I think it is important to refer here to Eq 44 indicating that for pure linear systems the balance part of the flow does not have vertical wind component.

    – **Authors' response**: Thank you.
    – **Authors' changes**: This has been done (Sect. 5.2). Note that Eq. (44) is now Eq. (45).

- **Referee comment**: Subsection 5.3 "Systematic exploration of model behaviour over parameter space" Figures 6, 8 and 9, panel d) The plots are very difficult to read, another line style/colours/symbols need to be used. Legends should be moved away because they destroy information on the plots.

    – **Authors' response**: Agreed.

    – **Authors' changes**: The text on these figures has been made larger, the figures have been made larger, and the balance diagnostic plots have been improved (the legends have been moved away from the lines, a colour scheme has been introduced, and a uniform scale has been used for all geostrophic and (separately) for all hydrostatic imbalances.

- **Referee comment**: Figure 7. It is difficult to read the plot. Another style/colours/symbols need to be used

    – **Authors' response**: Agreed.
    – **Authors' changes**: Change made (in addition to an added energy vs time plot for runs of the model with increased resolution).

- **Referee comment**: Technical Comments – p11 l16 . Should be "$\delta t = \Delta t/n$" instead of "$\delta t = \delta t/n$"

    – **Authors' response**: Agreed.
    – **Authors' changes**: Change made (Sect. 3.3.1).

- **Referee comment**: p15 l22. Should be "$\sigma_a$" instead of "$\sigma_g$"

    – **Authors' response**: Agreed.
    – **Authors' changes**: Change made (Sect. 4.5).

- **Referee comment**: p15 l24 . Should be "small-scale" instead of "scall-scale"

    – **Authors' response**: Agreed.
    – **Authors' changes**: Change made (Sect. 4.5).

---

## Author Comment (AC2) · 27 Aug 2017

[english]article        [T1]fontenc        [latin9]inputenc        geometry        verbose,tmargin=3cm,bmargin=3cm,lmargin=3.5cm,rmargin=3.5cm babel

[Figure]

**Response to referee 2**
We would like to thank referee 2 for his/her comments. The referee's comments that require attention to the paper are reproduced below preceded with "**Referee comment**", the authors' responses are preceded with "**Authors' response**", and our actions are preceded with "**Authors' changes**". We have produced a revised manuscript of the paper, but we understand that this is not meant to be uploaded with these comments. Only a brief explanation of the changes are given here and we hope that a revised manuscript will be requested where the detailed changes are given (in the text below we do refer to parts of the revised paper where any changes are made should a revised manuscript be requested).

- **Referee comment**: Having read the other review, I strongly agree with the request for a further analysis of the sample integrations in section 5, and their relation with the findings of the previous sections.

  – **Authors' response and changes**: please see report for reviewer 1.

- **Referee comment**: 1. Understandably, there are a lot of symbols. I believe it would be useful to have a list of symbols as an appendix. While reading the paper, I had to go back sometimes between sections to know the differences in variables, e.g. $p$ vs $p\_0$ vs $p\_{00}$, or when are the variables calligraphic, when do they have a star, etc.

  – **Authors' response**: Thank you, agreed.
– **Authors' changes**: A table has been added to the end of the introduction section.

- **Referee comment**: 2. It was a little difficult to read the axes in some of the figures. In fact, could some of the panels be done larger? For example, a figure has 4 panels stacked vertically with a lot of white space to the sides, while a 2x2 grid would show them better.

  – **Authors' response**: Figure 6 does have four panels stacked vertically as stated. We could make this into a 2x2 grid, but we would like to maintain the correspondence with Figs. 8 and 9 so that the $n$th row of each Fig. can be compared directly.
  – **Authors' changes**: Figures 6, 8 and 9 have been made larger and the bottom-most panels of each has been made clearer. Figure 7 has also been made larger and clearer (and now has two panels, to show how energy is numerically not conserved when the grid resolution is increased – to answer the other reviewer's comments).

- **Referee comment**: Typos: There are some typos both in the text and in the equations. For instance \delta t = \delta t /N. These have been identified by the other reviewer so I am not repeating them.

  – **Authors' response**: Thank you.
  – **Authors' changes**: Corrections made (please see report for reviewer 1).

---

## Author Response (AR2)

We would like to thank the editor and the reviewers for consideration of this manuscript.

The two corrections requested by the editor have been made (colour bar added to Fig. 4, and statement of sign of colours added to the caption of Fig. 5). In addition, the blue highlighting of the text of the previous submission has been removed.

We are very happy for the paper to be classified as "Development and technical paper" as suggested by the editor.

Ross Bannister (on behalf of all co-authors)